# Cost-effectiveness of habit-based advice for weight control versus usual care in general practice in the Ten Top Tips (10TT) trial: economic evaluation based on a randomised controlled trial

Nishma Patel,[1] Rebecca J Beeken,[2,3] Baptiste Leurent,[4] Rumana Z Omar,[5] Irwin Nazareth,[6] Stephen Morris[1]

[1]Department of Applied Health Research, University College London, London, UK
[2]Department of Behavioural Science and Health, University College London, London, UK
[3]Leeds Institute of Health Sciences, University of Leeds, Leeds, UK
[4]Department of Medical Statistics, London School of Hygiene & Tropical Medicine, London, UK
[5]Department of Statistical Science, University College London, London, UK
[6]Department of Primary Care and Population Health, University College London, London, UK

**Correspondence to**
Nishma Patel;
nishma.patel@ucl.ac.uk

## ABSTRACT

**Objective** Ten Top Tips (10TT) is a primary care-led behavioural intervention which aims to help adults reduce and manage their weight by following 10 weight loss tips. The intervention promotes habit formation to encourage long-term behavioural changes. The aim of this study was to estimate the cost-effectiveness of 10TT in general practice from the perspective of the UK National Health Service.

**Design** An economic evaluation was conducted alongside an individually randomised controlled trial.

**Setting** 14 general practitioner practices in England.

**Participants** All patients were aged ≥18 years, with body mass index ≥30 kg/m². A total of 537 patients were recruited; 270 received the usual care offered by their practices and 267 received the 10TT intervention.

**Outcomes measures** Health service use and quality-adjusted life years (QALYs) were measured over 2 years. Analysis was conducted in terms of incremental net monetary benefits (NMBs), using non-parametric bootstrapping and multiple imputation.

**Results** Over a 2-year time horizon, the mean costs and QALYs per patient in the 10TT group were £1889 (95% CI £1522 to £2566) and 1.51 (95% CI 1.44 to 1.58). The mean costs and QALYs for usual care were £1925 (95% CI £1599 to £2251) and 1.51 (95% CI 1.45 to 1.57), respectively. This generated a mean cost difference of −£36 (95% CI −£512 to £441) and a mean QALY difference of 0.001 (95% CI −0.080 to 0.082). The incremental NMB for 10TT versus usual care was £49 (95% CI −£1709 to £1800) at a maximum willingness to pay for a QALY of £20 000. 10TT had a 52% probability of being cost-effective at this threshold.

**Conclusions** Costs and QALYs for 10TT were not significantly different from usual care and therefore 10TT is as cost-effective as usual care. There was no evidence to recommend nor advice against offering 10TT to obese patients in general practices based on cost-effectiveness considerations.

**Trial registration number** ISRCTN16347068; Post-results.

### Strengths and limitations of this study

► The analysis is based on a large multicentre randomised trial with detailed information on resource use and utility values for a median follow-up period of 2 years.
► Individual data on standard weight-loss interventions received by the participant were not recorded.
► The analysis took a UK National Health Service (NHS)/personal social service perspective, but a wider perspective (eg, societal) could have been taken, including costs to patients and families.
► The time horizon was 2 years.
► Missing NHS resource use and utility data were accounted for using multiple imputation.

recent estimates in the European Union, overweight affects 30%–70% and obesity affects 10%–30% of adults.[1] In the UK 23.9% of women aged 18 years and over are obese—the highest proportion in Europe. The proportion of UK men who are obese is 22.1%, the second highest proportion in Europe behind Malta (24.7%).[2]

Health risks associated with obesity range from heart disease, stroke and type 2 diabetes, to all cancers, gall bladder disease and mortality.[3] It is estimated that health problems associated with being overweight or obese cost the National Health Service (NHS) in England more than £5 billion every year.[4]

Non-surgical interventions for weight loss are available and include dietary advice, physical activity and behaviour modification. These can be accessed mainly through primary care. Despite considerable investment in such interventions, there is limited evidence about their cost-effectiveness.[5–9]

Ten Top Tips (10TT) was a simple leaflet-based intervention for patients in primary

## INTRODUCTION

The worldwide prevalence of obesity nearly doubled between 1980 and 2008. Based on

care designed to help with weight loss. The underpinning theory behind 10TT was habit formation. Psychological research shows that repetition of a simple action in a consistent context leads, through associative learning, to the action being activated on subsequent exposure to those contextual cues.[10] A small pilot trial consisting of 100 people showed that those who received the 10TT leaflet lost significantly more weight than those who did not receive it.[11] Although the absolute weight loss was modest, the intervention had the potential to be disseminated at minimal cost to large numbers simply by making the leaflet available. In response to this a large randomised trial was undertaken. Participants were given a leaflet, which provided weight loss tips based on scientific evidence on how to turn the tips into healthy habits, during a 30 min appointment with a practice nurse. The 10 healthy tips included keeping to meal routines, reduced fat intake, walking off weight, packing a healthy snack, looking at food labels, taking caution with portion size, standing up, thinking about drinks, focusing on food and not forgetting 'five a day'. The 'five a day' campaign is based on the recommendation from the WHO encouraging consumption of at least 400 g of fruits and vegetables a day to lower the risk of serious health problems, such as heart disease, stroke and some cancers.[12] Participants were asked to record their progress in a logbook. Using routinely collected data from primary care practices, the aim of the present study was to analyse the cost and cost-effectiveness of 10TT.

## METHODS
### Trial background
The 10TT trial was a two-arm, individually randomised, controlled trial in which 537 obese men and women were enrolled.[13] Practices across England (n=14) were recruited through the Medical Research Council General Practice Research Framework. They were located in Wellingborough, Southampton, Bradford-on-Avon, Bromsgrove, Frome, Guisborough, Glastonbury, Ivybridge, Dunstable, Liskeard, Ledbury, New Mills and London. The majority were located in Southern England (n=9), with three located in the Midlands and two in the North. Recruitment occurred between August 2010 and October 2011. Participants were randomly assigned to 10TT (n=267) or usual care (n=270), and followed for up to 2 years. The primary outcome was weight loss at 3 months. The secondary outcomes included body mass index (BMI), waist circumference, the number of people achieving a 5% reduction in weight, clinical markers for potential comorbidities (blood pressure, total cholesterol/low-density lipoprotein and blood glucose) and maintenance of weight loss over 24 months. At 3 months participants in the 10TT group lost significantly more weight than those receiving usual care (mean difference in weight change=−0.87 kg, 95% CI −1.47 to −0.27, p=0.004). But this effect was not maintained at 24 months (mean difference in weight change=+0.75 kg, 95% CI −0.73 to 2.24). Weight loss in

the usual care group was slow in the first 6 months, but it continued until 18 months, whereas the 10TT group experienced a greater weight loss in the first 6 months, but did not lose any additional weight after this point.

### Patient and public involvement
There was patient representation on the trial steering group for 10TT; however, neither patients nor the public were involved in the economic analyses presented in this paper.

### Overview of economic evaluation
We undertook a cost-utility analysis to compare the costs and outcomes associated with 10TT and usual care. The outcome measure was quality-adjusted life year (QALY), which combines the length of life and quality of life, and is consistent with the National Institute for Health and Care Excellence (NICE) recommendations.[14] The primary outcomes were incremental costs and effects and the incremental net monetary benefit (NMB) of 10TT versus usual care. Costs were measured from an NHS and personal social services (PSS) perspective; since PSS costs were negligible, we focused on NHS costs. These were calculated in 2013/2014 UK£. The time horizon for costs and effects was 2 years, reflecting the trial follow-up period. Extrapolation beyond this time period was not undertaken because the within-trial analysis found no evidence of a significant difference in costs or outcomes at 24 months; 2 years was therefore considered long enough to reflect all important differences in costs or outcomes between treatments. Costs and outcomes in the second year were discounted at 3.5%.[14]

### Resource use and costs
Resource use data were extracted from general practitioner (GP) records for 2 years prior to randomisation and 2 years post randomisation on the number and type of contacts with the GP (surgery visit, home visit, phone call), practice nurse contacts (practice visit, home visit, phone call), dietitian visits, hospital outpatient visits, hospital inpatient stay, accident and emergency (A&E) visits, and outpatient services. Data on use of drugs included dose, type and frequency of administration. All resource use data were extracted by practice nurses at participating centres from patient records.

Unit costs were attached to each resource item. The cost per visit to primary care (GP surgery visit, GP home visit, GP phone call, practice nurse visit, nurse home visit, practice nurse phone call and dietitian visits) was taken from Personal Social Services Research Unit (PSSRU) estimates.[15] The costs of inpatient episodes, outpatient visits and A&E visits were taken from the PSSRU and based on the NHS reference costs.[16] They included costs of medical staff, equipment, consumables and diagnostics. The weighted average cost of inpatient stays was calculated combining the cost of inpatient long and short stays. Drug costs were obtained from the British National Formulary.[17]

The cost of the 10TT intervention included the following, valued using market prices: a logbook (£2.80 per participant); time patients spent with the practice nurse to introduce the programme (£20 for a 30 min visit); wallet-sized food label guidance (£0.09) and a 10TT leaflet (£0.05). The total cost of 10TT was £22.94 per participant. The cost of the food label guidance was based on the total cost of printing (£1392) divided by the number of labels printed (15 000). Similarly, the cost of 10TT leaflets was based on the total cost of printing (£1540) divided by the number of leaflets printed (30 000).

All practices offered standard weight-loss interventions as part of usual care. These consisted of a range of interventions including referrals to community programmes, gym prescription and referral to a dietitian and/or psychologist, among others. While data were provided by practices on interventions typically offered to their patients, the uptake of these interventions was not recorded. In our base case, we made the conservative assumption (possibly biased against 10TT) that no patients received these interventions. In a sensitivity analysis, we included the costs of these interventions assuming each patient in the control group received the standard weight-loss intervention offered by the practice. Where single interventions were offered by a practice, these were costed accordingly, based on the average cost of participating in Weight Watchers (assumed to cost £78.22 per patient[18]) or Slimming World (£72.62 per patient). Where multiple interventions were offered by a practice, these were costed based on the average cost of participating in the Size Down Programme at a cost of £93.48 per patient.[18]

Unit costs (total expenditure incurred by the NHS for one visit) for each cost component are shown in table 1. These were multiplied by resource volume and summed across all cost components to obtain a total cost per patient.

## Utilities and QALYs

Health utilities were based on the EuroQol 5-dimension 3-level (EQ-5D-3L) descriptive system.[19 20] This is a five-dimensional questionnaire (mobility, self-care, usual activities, pain and discomfort), with three levels in each dimension (severe problems, some problems, no problems). Each EQ-5D-3L state was converted into a single utility score based on valuations from the UK general population.[21] Utility values of 1 represent full health, values of 0 are equivalent to death, and negative values represent states worse than death. The EQ-5D-3L questionnaire was completed at baseline, 3 months, 6 months, 12 months, 18 months and 24 months. A utility profile was constructed for participants assuming a straight-line relation between their utility values at each measurement point. QALYs for every patient from baseline to 2 years were calculated as the area under the utility profile.

## Dealing with missing data

There were missing data for NHS resource use and utility scores. Multivariate imputation by chained equations was used to impute missing data.[22 23] The imputation method used an iterative Markov Chain Monte Carlo technique to simulate from the posterior predictive distribution of missing data.[24] We generated 20 imputed data sets. We imputed missing data (% of missing data) for the following variables: weight (0.2%); BMI (0.2%); waist circumference (0.6%); EQ-5D-3L at baseline (5%), 3 months (26%), 6 months (40%), 12 months (46%) and 24 months (46%); NHS visits (GP practice (28%), GP home (31%), GP phone calls (30%), practice nurse (27%), nurse home (31%), nurse phone calls (31%), extra nurse (30%), dietitian (31%), hospital inpatient stay (31%), outpatient clinic (30%), A&E visits (30%)); and other outpatient service visits (30%) (table 1). Age, sex and study centre were included in the imputation model as additional explanatory variables. Imputations were undertaken using the –mi impute mvn– command in Stata SE V.14.

## Statistical analysis

Two sample t-tests to test for differences in cost and QALYs between the two groups were carried out for complete data. A linear regression model was used to test for differences in mean resource use, costs, EQ-5D scores at each time point and QALYs[25 26] using the imputed data. The incremental NMBs were calculated as the difference in mean QALYs per participant ($Q$) with 10TT ($T$) versus usual care ($U$) multiplied by the maximum willingness to pay for a QALY ($R$) minus the difference in mean cost per participant ($C$), that is, incremental NMB=$(Q_T - Q_U)*R - (C_T - C_U)$. We used the cost-effectiveness threshold range recommended by NICE (£20 000–£30 000[14]) as the lower and upper limits of the maximum willingness to pay for a QALY ($R$). Negative incremental NMBs indicate that usual care is preferred on cost-effectiveness grounds, and positive incremental NMBs indicate that 10TT is preferred.[27]

We had initially adjusted for age, gender, practice and costs 2 years prior in the analysis for incremental costs. Similarly, we adjusted for age, gender, practice and baseline utility values in the analysis for incremental QALYs. There were no differences between the two groups in terms of these factors (table 2) and therefore an unadjusted model was used.

For each of the 20 imputed data sets, we ran 1000 bootstrap replications using non-parametric bootstrapping, resampling observations with replacement.[28] The bootstrapped results were combined using the formula described by Briggs et al,[23] to calculate the mean values for costs and utilities and the SEs around the imputed values. SEs were based on a normal distribution and used to calculate 95% CIs around point estimates.

The cost-effectiveness plane was used to illustrate the difference in costs and outcomes between the two groups. The probability that 10TT was cost-effective in comparison

**Table 1** Resource use, unit cost and mean cost per participant for primary and secondary care services postrandomisation

| | | Usual care | | 10TT | | | Usual care | 10TT | Incremental difference | |
|---|---|---|---|---|---|---|---|---|---|---|
| | n | Mean (SD) | Median (IQR) | Mean (SD) | Median (IQR) | Unit cost (£) | Mean cost (£) | Mean cost (£) | Mean (£) | (95% CI) |
| No imputation for missing values | | | | | | | | | | |
| GP surgery visit | 204 | 9.4 (8.4) | 7 (7.00) | 9.7 (9.5) | 7 (9.00) | 45 | 417 | 425 | 8 | (−72 to 89) |
| GP home visit | 196 | 0.1 (0.8) | 0 (0.00) | 0.0 (0.1) | 0 (0.00) | 292 | 37 | 9 | −27 | (−61 to 7) |
| GP phone | 199 | 1.3 (3.0) | 0 (1.00) | 1.3 (3.0) | 0 (1.00) | 27 | 34 | 34 | 0 | (−16 to 16) |
| GP practice nurse | 206 | 6.9 (8.4) | 4 (6.00) | 6.8 (9.5) | 4 (6.50) | 40 | 274 | 268 | −6 | (−77 to 64) |
| Nurse home visit | 195 | 0.1 (1.1) | 0 (0.00) | 0.0 (0.1) | 0 (0.00) | 70 | 5 | 1 | −4 | (−16 to 7) |
| GP practice nurse phone | 197 | 0.5 (2.3) | 0 (0.00) | 0.4 (1.2) | 0 (0.00) | 10 | 5 | 4 | −1 | (−5 to 2) |
| Dietitian | 195 | 0.1 (0.5) | 0 (0.00) | 0.1 (0.8) | 0 (0.00) | 35 | 4 | 4 | 0 | (−4 to 4) |
| Additional nurse visit | 197 | 1.1 (3.1) | 0 (1.00) | 1.6 (2.6) | 0 (1.00) | 4 | 45 | 44 | 0 | (−23 to 22) |
| Inpatient admission | 196 | 0.2 (0.7) | 0 (0.00) | 0.2 (0.7) | 0 (0.00) | 1713 | 372 | 387 | 14 | (−220 to 249) |
| Outpatient clinic | 200 | 2.2 (3.5) | 1 (3.00) | 2.1 (3.3) | 1 (3.00) | 135 | 288 | 269 | −20 | (−110 to 70) |
| Accident and emergency visit | 199 | 0.4 (1.2) | 0 (0.00) | 0.4 (0.9) | 0 (1.00) | 177 | 65 | 67 | 2 | (−36 to 39) |
| Other outpatient services | 194 | 1.0 (1.8) | 0 (1.00) | 1.0 (2.3) | 0 (1.00) | 135 | 124 | 129 | 5 | (−48 to 58) |
| Missing values imputed* | | SE | | SE | | | | | | |
| GP surgery visit | 270 | 9.4 (0.6) | 9 | 9.4 (0.7) | 9 | 45 | 419 | 420 | 1 | (−79 to 81) |
| GP home visit | 270 | 0.1 (0.5) | 0 | 0.0 (0.2) | 0 | 292 | 36 | 13 | −23 | (−58 to 12) |
| GP phone | 270 | 1.3 (0.2) | 1 | 1.3 (0.2) | 1 | 27 | 35 | 33 | −2 | (−19 to 14) |
| GP practice nurse | 270 | 6.9 (0.6) | 6 | 6.6 (0.7) | 6 | 40 | 274 | 266 | −9 | (−75 to 57) |
| Nurse home visit | 270 | 0.1 (1.0) | 0 | 0.0 (0.1) | 0 | 70 | 6 | 2 | −4 | (−16 to 7) |
| GP practice nurse phone | 270 | 0.5 (0.6) | 0 | 0.4 (0.1) | 0 | 10 | 5 | 4 | −1 | (−5 to 2) |
| Dietitian | 270 | 0.1 (0.0) | 0 | 0.1 (0.6) | 0 | 35 | 4 | 4 | 0 | (−4 to 4) |
| Additional nurse visit | 270 | 1.2 (0.2) | 1 | 1.8 (0.2) | 0 | 4 | 46 | 47 | 0 | (−23 to 23) |
| Inpatient admission | 270 | 0.2 (0.1) | 0 | 0.2 (0.1) | 0 | 1713 | 373 | 379 | 6 | (−238 to 251) |
| Outpatient clinic | 270 | 2.1 (0.3) | 2 | 2.0 (0.2) | 0 | 135 | 287 | 266 | −22 | (−106 to 63) |
| Accident and emergency visit | 270 | 0.4 (0.9) | 0 | 0.4 (0.7) | 0 | 177 | 67 | 69 | 2 | (−38 to 42) |
| Other outpatient services | 270 | 1 (0.1) | 1 | 1 (0.2) | 0 | 135 | 122 | 129 | 6 | (−52 to 65) |
| Weight-loss interventions | | | | | | | | | | |
| 10TT | | – | – | – | – | 23 | – | – | | |
| Weight Watchers | | – | – | – | – | 78 | – | – | | |
| Slimming World | | – | – | – | – | 93 | – | – | | |
| Size Down Programme | | – | – | – | – | 73 | – | – | | |

Costs are in 2013/2014 UK£.
SD for non-imputed data. SE for imputed data.
*Missing values imputed using multiple imputation with 20 imputed data sets (see text).
10TT, Ten Top Tips; GP, general practitioner.

with usual care at different values of the cost-effectiveness threshold was presented in the form of a cost-effectiveness acceptability curve (CEAC).[29] The CEAC was based on the proportion of bootstrap replications across all 20 imputed data sets that were below the cost-effectiveness threshold, which was varied from £0 to £50 000.

Several deterministic sensitivity analyses were carried out to assess the uncertainty around key components of the analysis. An analysis was undertaken based on complete cases without any imputation. We varied cost components

for primary and secondary care costs and drug costs. The total costs of each component were amended by ±10%. We also conducted an analysis including the cost of standard weight-loss interventions for the usual care group.

## RESULTS
### Baseline description
Table 3 illustrates the mean costs 2 years prior to the trial. The mean cost for the usual care group was £1848

**Table 2** Demographics at baseline

| | Usual care | | | 10TT | | | |
|---|---|---|---|---|---|---|---|
| | n | Mean* | SD | n | Mean | SD | P values† |
| Age | 270 | 58 | 12.61 | 267 | 57 | 12.88 | 0.47 |
| Male | 95 | 35.2 | – | 89 | 33.3 | – | – |
| Female | 175 | 64.8 | – | 178 | 66.7 | – | – |
| Prior costs | 270 | 1848 | 1948 | 267 | 2052 | 2461 | 0.29 |
| Weight (kg) | 269 | 101 | 17.46 | 267 | 100 | 16.98 | 0.59 |
| Body mass index | 269 | 36.59 | 5.72 | 267 | 36.18 | 4.71 | 0.36 |
| Baseline EQ-5D-3L | 257 | 0.76 | 0.24 | 255 | 0.74 | 0.27 | 0.41 |

*Figures for sex based on the mean proportion (%) in each group.
†From independent t-test.
10TT, Ten Top Tips.
EQ-5D-3L, EuroQol 5-dimension 3-level descriptive system.

(95% CI 1615 to 2082) and £2052 (95% CI 1756 to 2349) for the 10TT group. Baseline characteristics—age, gender, weight, BMI, blood pressure and cholesterol—did not differ between groups.[13]

### Base case

Table 1 shows the mean imputed resource use by treatment group. The intervention and control groups had the same mean GP practice visits per participant over the 2-year period (9.4 vs 9.4), as well as similar GP home visits (0.0 vs 0.1), GP phone contacts (1.3 vs 1.3), nurse practice visits (6.6 vs 6.9), nurse home visits (0.0 vs 0.1), inpatient episodes (0.2 vs 0.2), outpatient clinic visits (2.0 vs 2.1) and A&E visits (0.4 vs 0.4). There was no statistically significant difference between groups for any category of health service resource use.

Table 3 shows the total cost per participant in each group. The non-parametric bootstrapping including imputed resource use produced a mean total cost per participant in the intervention group of £1889 (95% CI £1522 to £2566), compared with £1925 (95% CI £1599 to

£2251) in the control group. The mean difference in cost between 10TT and control group was –£36 (95% CI –£512 to £441), which was not statistically significant (p=0.88).

Undertaking non-parametric bootstrapping after multiple imputation produced 1.51 (95% CI 1.44 to 1.58) QALYs in the intervention group and 1.51 (95% CI 1.45 to 1.57) in the control group, generating a mean difference in QALYs of 0.001 (95% CI –0.080 to 0.082) (p=0.93) (tables 4 and 5), which was not statistically significant. Hence, patients receiving 10TT accrued non-significantly lower costs, and the difference in QALYs between the groups was very small and non-significant.

The incremental NMB for 10TT versus usual care was £49 (95% CI –£1709 to £1800) at a maximum willingness to pay for a QALY of £20 000 and £55 (95% CI –£2489 to £2583) at a maximum willingness to pay for a QALY of £30 000 (table 4).

Of the 20 000 bootstrap replications produced under base case assumptions, 24% fell into the south-west quadrant of the cost-effectiveness plane (10TT was less costly

**Table 3** Cost description 2 years prior and post randomisation

| | Usual care | | | 10TT | | | Incremental difference |
|---|---|---|---|---|---|---|---|
| | n | Mean | SD | n | Mean | SD | Mean (£) (95% CI) |
| Costs 2 years prior | | | | | | | |
| Cost of primary and secondary care contacts | 270 | 1516 | 1694 | 267 | 1735 | 2186 | 219 (–112 to 550) |
| Drugs costs | 270 | 332 | 748 | 267 | 317 | 624 | –15 (–132 to 102) |
| Total cost | 270 | 1848 | 1948 | 267 | 2052 | 2461 | 204 (–172 to 580) |
| Costs after randomisation (imputed)* | | SE | | | SE | | |
| Cost of primary and secondary care contacts | 270 | 1675 | 150 | 267 | 1631 | 169 | –45 (–477 to 387) |
| Drugs costs | 270 | 249 | 44 | 267 | 236 | 68 | –13 (–172 to 146) |
| Intervention costs | – | – | – | 267 | 23 | – | – |
| Total cost | 270 | 1925 | 165 | 267 | 1889 | 186 | –36 (–512 to 441) |

SD for non-imputed data. SE for imputed data.
*Missing values imputed using multiple imputation with 20 imputed datasets (see text).
10TT, Ten Top Tips.

**Table 4** Incremental cost-effectiveness of 10TT versus usual care

| | Incremental cost | | Incremental QALYs | | Incremental net monetary benefit | | | | Probability of being cost-effective (%) | |
| --- | --- | --- | --- | --- | --- | --- | --- | --- | --- | --- |
| | | | | | £20000 | | £30000 | | | |
| | Mean (£) | (95% CI) | Mean | (95% CI) | Mean (£) | (95% CI) | Mean (£) | (95% CI) | £20000 | £30000 |
| Base case* | −36 | (−512 to 441) | 0.001 | (−0.080 to 0.082) | 49 | (−1709 to 1800) | 55 | (−2489 to 2583) | 52 | 52 |
| Base case with control group intervention cost† | −122 | (−598 to 353) | 0.001 | (−0.080 to 0.082) | 140 | (−1666 to 1902) | 148 | (−2463 to 2693) | 56 | 54 |
| Complete case analysis‡ | −66 | (−907 to 774) | −0.047 | (−0.180 to 0.086) | −889 | (−3993 to 2253) | −1361 | (−5772 to 3052) | 29 | 28 |
| Primary and secondary care costs increased§ | −40 | (−556 to 475) | 0.001 | (−0.080 to 0.081) | 48 | (−1743 to 1851) | 53 | (−2520 to 2623) | 52 | 51 |
| Drug costs increased§ | 23 | (−396 to 442) | 0.001 | (−0.080 to 0.077) | −14 | (−1734 to 1716) | −9 | (−2511 to 2510) | 49 | 50 |
| All costs increased§ | −39 | (−554 to 476) | 0.001 | (−0.080 to 0.082) | 43 | (−1756 to 1851) | 46 | (−2537 to 2641) | 52 | 51 |
| Primary and secondary care costs decreased¶ | −31 | (−463 to 401) | 0.001 | (−0.080 to 0.081) | 54 | (−1683 to 1804) | 64 | (−2453 to 2587) | 52 | 52 |
| Drug costs decreased¶ | 26 | (−387 to 438) | 0.001 | (−0.080 to 0.081) | 0 | (−1741 to 1727) | 11 | (−2530 to 2520) | 50 | 51 |
| All costs decreased¶ | −32 | (−460 to 395) | 0.001 | (−0.080 to 0.081) | 50 | (−1682 to 1811) | 59 | (−2458 to 2599) | 52 | 52 |

*n=537. Data include values imputed using multiple imputation (see text). The 95% CIs were derived from 1000 bootstrap replications of each of the 20 imputed data sets (see text). The incremental net monetary benefit and the probability 10TT is cost-effective are based on the QALYs gained and incremental costs and calculated at a maximum willingness to pay for a QALY of £20000 and £30000.

†As for the base case analysis. Standard weight-loss interventions (Weight Watchers, Slimming World, Size Down Programme) are included in the control group.

‡As for the base case analysis. No multiple imputation of missing values and the 95% CIs were derived from 1000 bootstrap replications of a single data set containing the n=72 participants in the 10TT group and n=91 participants in the usual care group with no missing values. The costs of interventions in the control group (Weight Watchers, Slimming World, Size Down Programme) were excluded.

§As for the base case analysis. Costs were increased by 10% for 10TT.

¶As for the base case analysis. Costs were decreased by 10% for 10TT.

10TT, Ten Top Tips; QALYs, quality-adjusted life years.

**Table 5** Quality-adjusted life years (QALYs) per patient

| | Usual care | | | | 10TT | | | | Incremental difference | |
|---|---|---|---|---|---|---|---|---|---|---|
| | n | Mean | SD | (95% CI) | n | Mean | SD | (95% CI) | Mean | (95% CI) |
| No imputation for missing values | | | | | | | | | | |
| Utility at 3 months | 206 | 0.77 | 0.23 | (0.74 to 0.80) | 190 | 0.77 | 0.26 | (0.73 to 0.80) | 0.00 | (−0.05 to 0.05) |
| Utility at 6 months | 163 | 0.76 | 0.24 | (0.72 to 0.80) | 159 | 0.78 | 0.24 | (0.74 to 0.82) | 0.02 | (−0.03 to 0.07) |
| Utility at 12 months | 151 | 0.77 | 0.24 | (0.73 to 0.81) | 138 | 0.74 | 0.27 | (0.70 to 0.79) | −0.03 | (−0.09 to 0.03) |
| Utility at 18 months | 138 | 0.75 | 0.25 | (0.71 to 0.79) | 124 | 0.74 | 0.27 | (0.69 to 0.79) | −0.01 | (−0.07 to 0.05) |
| Utility at 24 months | 157 | 0.76 | 0.25 | (0.73 to 0.80) | 131 | 0.76 | 0.26 | (0.72 to 0.81) | 0.00 | (−0.06 to 0.06) |
| Discounted QALYs (24 months) | 92 | 1.52 | 0.40 | (1.44 to 1.61) | 73 | 1.47 | 0.47 | (1.36 to 1.58) | −0.05 | (−0.18 to 0.08) |
| Missing values imputed* | | SE | | | | SE | | | | |
| Utility at 3 months | 270 | 0.77 | 0.016 | (0.73 to 0.80) | 267 | 0.77 | 0.017 | (0.73 to 0.80) | 0.00 | (−0.05 to 0.05) |
| Utility at 6 months | 270 | 0.76 | 0.016 | (0.73 to 0.79) | 267 | 0.78 | 0.018 | (0.73 to 0.81) | 0.01 | (−0.03 to 0.06) |
| Utility at 12 months | 270 | 0.76 | 0.019 | (0.73 to 0.80) | 267 | 0.75 | 0.020 | (0.71 to 0.80) | −0.01 | (−0.06 to 0.04) |
| Utility at 18 months | 270 | 0.76 | 0.019 | (0.72 to 0.79) | 267 | 0.75 | 0.020 | (0.71 to 0.79) | −0.01 | (−0.05 to 0.04) |
| Utility at 24 months | 270 | 0.76 | 0.019 | (0.72 to 0.79) | 267 | 0.76 | 0.018 | (0.72 to 0.79) | 0.00 | (−0.05 to 0.05) |
| Discounted QALYs (24 months) | 270 | 1.51 | 0.030 | (1.45 to 1.57) | 267 | 1.51 | 0.033 | (1.44 to 1.58) | 0.00 | (−0.08 to 0.08) |

Costs are in 2013/2014 UK£. CIs were based on non-parametric bootstrapping for observed data and imputed data.
SD for non-imputed data. SE for imputed data.
*Missing values imputed using multiple imputation with 20 imputed data sets (see text).
10TT, Ten Top Tips.

and less effective than usual care); 20% fell into the northeast quadrant (10TT was more costly and more effective than usual care); 25% fell into the north-west quadrant (10TT was more costly and less effective than usual care); and 31% fell into the south-east quadrant (10TT was less costly and more effective than usual care) (figure 1). The CEAC derived from figure 1 is illustrated in figure 2. Under base case assumptions, 10TT has a 52% probability of being cost-effective at a willingness to pay threshold of £20 000 and 52% at a willingness to pay threshold of £30 000 (table 4).

### Sensitivity analyses
Including the cost of the standard weight-loss interventions (£87) in the usual care group, the mean costs per patient in the control group increased to £2012, which was higher than the mean cost in the intervention group (£1889). The mean incremental cost difference between groups was −£122 (95% CI −£598 to £353). The incremental NMB for 10TT versus usual care was £140 (95% CI −£1666 to £1902) at a maximum willingness to pay for a QALY of £20 000 and £148 (95% CI −£2463 to £2693) at a maximum willingness to pay of £30 000. Increasing

and decreasing costs by 10% did not affect the findings appreciably (table 4).

A complete case analysis was undertaken using 163 of 537 participants with complete utility and cost data. The incremental NMB was −£889 (95% CI −£3993 to £2253) at a maximum willingness to pay for a QALY of £20 000 and −£1361 (95% CI −£5772 to £3052) at a maximum willingness to pay for a QALY of £30 000.

### DISCUSSION
Our economic analysis of 10TT showed that this intervention had similar costs and QALYs as usual care. Sensitivity analyses showed little uncertainty in this finding. On the one hand the findings mean there is no reason to prefer 10TT or usual care on the basis of differences in quality of life or cost, or on cost-effectiveness grounds. On the other hand, this means that 10TT is as cost-effective as usual care.

We undertook a rapid review to compare our results with similar weight loss programmes. We found that commercial weight loss programmes were highly

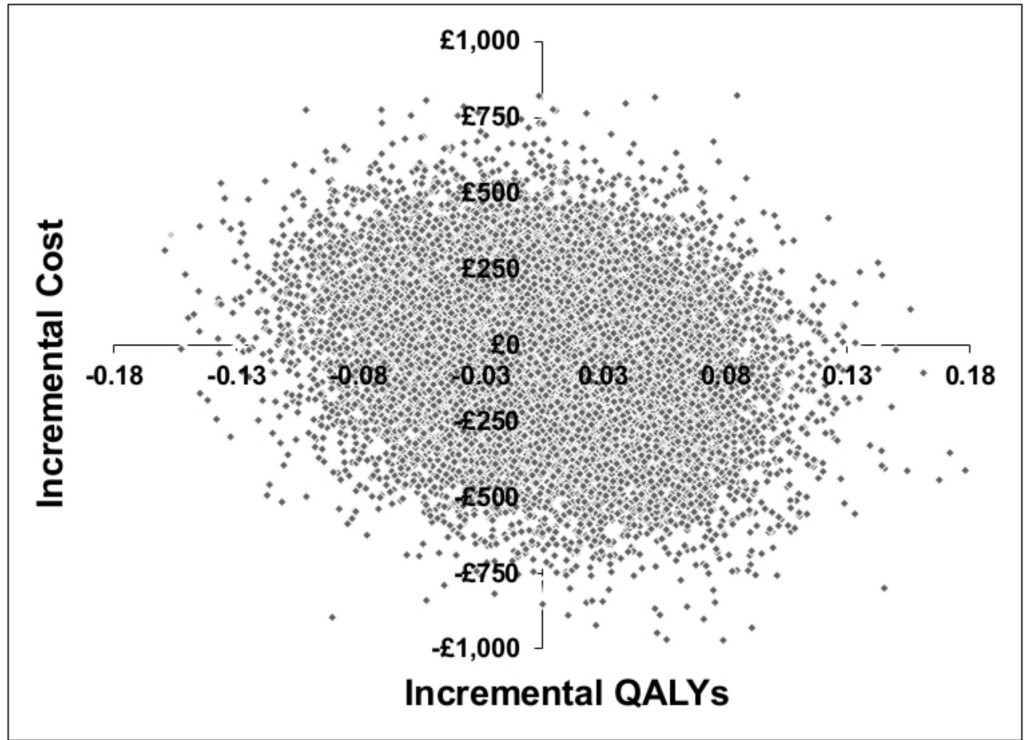

**Figure 1** Cost-effectiveness plane, base case analysis assumptions. Based on 20 000 bootstrap replications, from 20 imputed data sets.

prescribed among primary care providers, and these participants lost more weight than those in self-led education programmes alone. Fuller et al[7] reported the long-term analysis of a 20 min GP consultation versus Weight Watchers and found Weight Watchers produced a cost saving of US$47 per patient and an incremental 0.03 QALY gained per patient. Similarly, a recent evaluation in the UK of a primary care-led behavioural programme[30] looked at a brief advice and self-help materials (primary care-led programme) versus Weight Watchers over 12 weeks and over 52 weeks. The authors concluded that Weight Watchers was more effective over 12 weeks (−4.75 kg) and 52 weeks (−6.76 kg) than brief advice and self-help material (−3.26 kg), at a cost of £159 per kilogram lost. Additionally, a primary care-led programme

Counterweight (a nurse-delivered patient education programme) showed that nurse-delivered education was less costly and more effective compared with no active intervention, producing a gain in QALYs (0.06 per participant) and cost savings of £27 per participant.[9]

It is evident from existing literature that GPs play a crucial role in obesity prevention and weight management and are gatekeepers to lifestyle weight management programmes.[31] While there is evidence to suggest GP prescribed commercial programmes and/or weight-loss education is effective, further research is needed to explore the relationship between habit formation programmes such as 10TT and commercial programmes, with the aim to determine what the long-term cost savings and QALYs produced would potentially be over a long-time horizon.

The main strength of our analysis is that it is based on a large multicentre randomised trial with detailed information on resource use and utility values for a median follow-up period of 2 years. There were several limitations to our study. While data were recorded in the trial on standard weight-loss interventions for obese patients at each practice, individual data on the uptake and prescribing of standard weight-loss interventions received by the participant were not recorded. This trial was also unblinded, which introduced the potential bias of GPs over prescribing standard weight-loss interventions producing a healthy user bias effect. Under base case assumptions, excluding standard weight-loss interventions, the incremental NMB of 10TT versus usual care

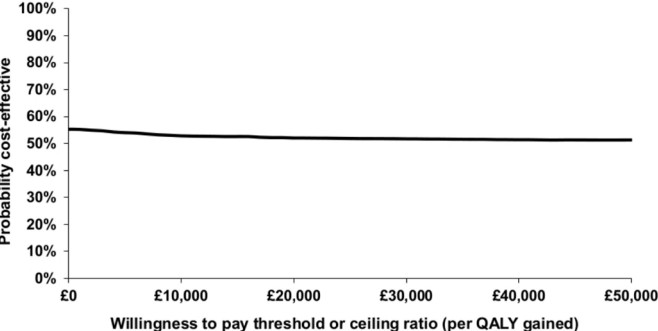

**Figure 2** Cost-effectiveness acceptability curve showing the probability that Ten Top Tips is cost-effective versus usual care at different values of the maximum willingness to pay for a quality-adjusted life year (QALY) (base case assumptions).

was not significantly different from zero, and this did not change when intervention costs in the control group were included. Hence, these costs did not make an appreciable difference to our conclusions. Second, the analysis took a UK NHS/PSS perspective. A wider perspective (eg, societal) could have been taken, including costs to patients and families. Third, the time horizon was limited to 2 years due to the lack of difference in costs (–£36, 95% CI –£512 to £441, p value 0.88) and benefits (0.001, 95% CI –0.080 to 0.082, p value 0.93) at 24 months between groups. The trial did not capture QALYs from weight loss post 24 months. A longer follow-up period may have allowed for the detection of long-term health benefits produced as a consequence of avoiding obesity-related health conditions, such as diabetes and cardiovascular disease. However, at the time of this study little was known about the association between habit formation and weight loss. This study has identified the importance of longer term strategies for continued adherence of weight loss. With add-on approaches such as counselling and education on how to maintain weight loss for participants alongside 10TT, it may be possible to maintain weight loss post 2 years.

Fourth, while there are various instruments available to measure health-related quality of life, we administered the EQ-5D.[32] We acknowledge that there may have been a potential value in using more than one measure in the trial, such as the Short Form Health Survey (SF-36) and the Impact of Weight on Quality of Life-Lite to measure differences in the positive short-term psychological effects.[33] However, given the small difference between the two groups, it is highly unlikely alternative measures would have produced significant differences. Finally, we were unable to access Hospital Episode Statistics (HES) data containing detailed secondary care resource use of NHS services by patients. Obtaining HES data would have been problematic as these data would need to be linked to HES data by patient ID. Given the short time frame of this trial, this was not feasible.[34] Where data were available for secondary care and missing (inpatient admissions, A&E visits, outpatient clinic and other outpatient services), this was accounted for using multiple imputation, assuming these data were missing at random. It is important to note that when using such methods there is uncertainty around the non-observed value across the imputations. To account for the uncertainty around the values, we employed the non-parametric bootstrap approach to estimate the variance (a representation of uncertainty) around the true values.[23] We acknowledge that although multiple imputation is able to statistically test for error, this method can produce bias. The bias arises from the assumption that missing data in the study were 'missing at random'. For example, the missing at random assumption may be reasonable if a variable that is predictive of missing data is included in the imputation model, but not if the variable is omitted from the model.[35] However, multiple imputation offers an advantage over complete case analysis, by filling in missing values and enabling participants to be included in the analysis, where there were auxiliary variables that are strong predictors of missingness.[36] Guidelines were followed to ensure the model predicted the best possible estimates: investigation of the proportion of missing data; selection of specific variables; and identification of the predictors of missingness.[37] We acknowledge that as the percentage of missingness increases (in some cases more than 50%), there will be greater inefficiency in the imputed data and more chance of bias. However, in general the application of regression imputation with large missing data may be acceptable if adjustments are made for the predictions, as we have done.[23]

In conclusion, the 10TT trial found that after 3 months participants receiving 10TT lost significantly more weight than those receiving usual care, but there was no difference in weight change between the two groups at 24 months. Similarly, there was no significant difference in costs and QALYs between the two arms at 24 months. There is no evidence to either recommend or advise against 10TT, based on cost-effectiveness considerations.

**Acknowledgements** Weight Concern and Cancer Research UK developed the 10 Top Tips intervention and materials, and Weight Concern provided training on how to deliver the intervention. The General Practice Research Framework provided advice on the study design and was involved in both recruiting the sites and training the health professionals at each site on the trial procedures, along with providing quality control visits. Practices were located in Wellingborough, Southampton, Bradford-on-Avon, Bromsgrove, Frome, Guisborough, Glastonbury, Ivybridge, Dunstable, Liskeard, Ledbury, New Mills and London. The authors would like to acknowledge the substantial intellectual contribution made by Professor Jane Wardle, who sadly passed away prior to publication and is deeply missed by all of her coauthors, colleagues and students.

**Contributors** NP undertook data analysis, interpreted the results, and drafted and wrote the manuscript. RJB helped design the study, undertook data collection and critically reviewed the manuscript. BL helped design the study, and critically reviewed the data and the manuscript. RZO helped design the study and critically reviewed the manuscript. IN helped design the study and critically reviewed the manuscript. SM undertook the data analysis, devised the analysis plan, interpreted the results and critically reviewed the manuscript.

**Funding** This work was funded by the Medical Research Council (MRC) - National Prevention Research Initiative (NPRI) (UK) grant number G080202. RJB's post was funded through this grant and she is now supported by Yorkshire Cancer Research Academic Fellowship funding. SM, NP and IN are HEFCE-funded. RZO was funded by the National Institute for Health Research.

**Disclaimer** The views expressed are those of the authors and not necessarily those of the funding bodies.

**Competing interests** None declared.

**Patient consent** Obtained.

**Ethics approval** This study has been approved by the South East London REC 2, 23/08/2010 (reference number 10/H0802/59). All participants gave informed consent before taking part.

**Provenance and peer review** Not commissioned; externally peer reviewed.

**Data sharing statement** No additional data are available.

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
