## [Reviewer comments · BMJ Open]

ARTICLE DETAILS

TITLE (PROVISIONAL)	Cost-effectiveness of habit-based advice for weight control versus usual care in general practice in the Ten Top Tips (10TT) Trial: economic evaluation based on a randomised controlled trial
AUTHORS	Patel, Nishma; Beeken, Rebecca; Leurent, Baptiste; Omar, Rumana; Nazareth, Irwin; Morris, Stephen

VERSION 1 – REVIEW

REVIEWER	Dr Ed Wilson University of Cambridge, UK
REVIEW RETURNED	27-May-2017

GENERAL COMMENTS	General Overall this appears to have been a well conducted and reported economic evaluation alongside a clinical trial. My main concern is with the use of EQ5D based QALYs over a period of only two years in a weight-loss trial. Whilst there may be some short-term positive psychological effect of weight loss (eg Peckmezian & Hay J Eat Disord 2017 PMID: 28489914) which may be reflected in the EQ5D, the major impacts will be long term from avoidance of T2DM, arthritis and some cancers and their associated complications. Therefore, a 2-year time horizon will not capture these. No decision modelling was undertaken due to the lack of a significant treatment effect at 24 months (although in any case I note there were no plans to do so noted in the protocol). As per my comments below I think it is still important to report the mean difference and 95%CI as this would be useful for future meta-analysis. My preference would be to see some longer-term decision modelling based on the 24 month data, so that parameter uncertainty is translated into decision uncertainty, but I suspect the (probably small) added value to decision makers would not justify the effort. Perhaps the best way to handle this comment is to add a few sentences to the discussion. Specific P4L75 – avoid imprecise, emotive terms such as ‘huge’: be specific. Perhaps simply delete the first sentence of the paragraph. P4L76-77 “Health risks associated with obesity...” – this claim needs a reference (eg Guh et al. BMC Pub Health 2009 Doi: 10.1186/1471-2458-9-88) P4L78-80 “as the prevalence...”: The NOO chart appears to show a levelling off of obesity prevalence since around 2008 – the rise is certainly a lot slower than between say 1994 and 2003.
---

	P4L81 – P6L112: Comparison with other papers should be merged with the respective points in the discussion. A shorter concise introduction is better – perhaps just state that non-surgical interventions exist but there is little evidence on cost-effectiveness, and cite the papers but save the actual numbers for the discussion. A new paper of relevance to this is Ahern et al. Lancet 2017 PMID:28478041 P6L126: Non-UK readers may not be familiar with the ‘five-a-day’ campaign. Suggest you provide a few words explaining. P7L134-137: It may be more informative to international readers to simply state the regions of England where recruitment took place and name the locations in the acknowledgements. P7L144-145: please state the point estimate and 95%CI around the 24m outcome, rather than just that there is ‘no [significant] difference’. P8L158-160 ‘found no evidence of significant differences’: I am very wary of this line of reasoning – absence of proof is not proof of absence, and the definition (p-value cut-off) of what is statistically significant is in any case utterly arbitrary. Apart possibly from a boost to mental wellbeing, I would not expect the EQ5D to show any effect whatsoever from weight loss over 2 years. Likewise, with NHS costs – the major benefits will be in avoiding T2DM and its complications in 10+ years, so a 2 year time horizon will not capture these events. A preferable strategy would be to input the observed differences – and uncertainty - in weight loss at 2 years into a model and translate that parameter uncertainty through to decision uncertainty. The counter to this would be if there really was zero effect of the intervention by 2 years (proof of absence), and that the period of lower weight had no cumulative effect on future risk of obesity related disease. (This would be in contrast to T2DM, where I think there is some evidence that a period of tight blood glucose control in T2DM still has beneficial effects even if control subsequently deteriorates, presumably through a reduced accumulation of damage – from memory one of the UKPDS papers reported on this). Reporting the point estimate and 95%CI from the non-sig difference at 12 months would be helpful in justifying a decision not to model the longer term costs and outcomes. P8L164: typo: post or prior to randomisation? (same typo in Table 4?) Were any data extracted prior to randomisation (eg 3m prior to) as a ‘baseline’ cost to adjust for in regression analysis? (The later text suggests data were extracted 2yrs prior to 2 years post randomisation). P8L166-167: Primary care records are notoriously bad at recording secondary care data. Why did you not access HES for these? (Too late to do anything about this now(!), but perhaps note this in the discussion). P10: Missing data: Please state the proportion of missing data for each parameter. P11L236: Is reference 21 correct? This appears to be a CUA, not a methods paper on INB. A better reference would be Zethraeus et al. Pharmacoeconomics 2003. Doi: 10.2165/00019053-200321010-
--	--

	00003 P11L237-242: Utility, and therefore probably QALYs too, are usually left skewed, especially in a mostly healthy population. Would a GLM have provided a better fit to the data than OLS in this case? Tables 2-4: I would strongly advise dropping the p-values from Tables 2-4. It would be better to report 95% CIs around differences, even better to limit increments to just total cost, or by sector too. Minor: P6L125: ?typo 'healthy' or 'health' snack? Throughout: Data are plural, not singular.
--	---

REVIEWER	Fredrik Norström Umeå University, Sweden
REVIEW RETURNED	09-Jun-2017

GENERAL COMMENTS	The article is very well written and addresses an important topic. Interestingly the study does not come up with the results that might have been expected from the authors, i.e. that the weight reduction intervention clearly is cost-effective. The result of the cost-effectiveness analysis, which to some extent could be called "negative" results should be important contribution to the field. The result indicates that only giving cost advices, at least in a similar way as described in the article, is not sufficient to have an effective weight reduction in the long run. All of the revisions I request below are of rather minor character. Methods:  1) Row 162: According to the CHEERS (item 9) statement should the discount rate be backed up by reference or similar. 2) Row 171-173: Missing end parenthesis. 3) Row 202: Specify that you use EuroQol 5D on first occasion instead of starting with the short form of the instrument. 4) Row 215: "used" should likely be "was used". 5) Row 225-226: A reference is required for the use of "A linear regression model for imputed data" as this is a non-standard method. 6) Row 227-228: It is unclear how and why non-parametric bootstrap has been used to calculate the difference in mean costs. It seems like the purpose with the bootstrap technique is to estimate the uncertainty (regardless if confidence intervals or graphical illustrations are the target for this) and not the mean costs. If so, revise the text accordingly. Bootstrap is not a well-known technique for any researcher so I recommend to add a reference for your method. 7) Row 237-238: The description needs to be improved. 8) Row 247-248: I assume that the CIs are calculated based on normality as you use the standard errors but this is not specified. As you use Bootstrap estimates elsewhere would this information be valuable to make it clear. 9) It is mentioned that you use CEAC but not cost-effectiveness planes in the method. Add such information. Tables:  1) Table 1: CI should be explained with confidence interval in the footnotes as should SE be. The general recommendation is to
---

	present SD instead of SE as it is not sensitive to the sample size, thus enables numerical comparisons of potential differences in variation between results so your manuscript might benefit from such change. 2) Table 2: SD = standard deviation is mentioned despite SE used in the table. 3) Table 4: The title is too unspecific in relation to the content of the table. 4) Table 5: It is not clear when imputed values have been used and not for the analyses in the table. References: 1) Reference 7 is a World Bank document, while the reference refer to results from a weight loss program. This needs to be corrected. 2) Reference 20 does not fit on row 206. 3) Reference 19 and 20 are identical. Please correct this. 4) Check if both references 21 and 22 are valid as references for how to use multiple imputation which they both are indicating now. 5) Row 223: References need to be fixed. Journal also likely to require city, state and country for the software here. 6) Author name (P R) for reference 23 is not complete. 7) Check also other references to see if they are correctly specified as there are so many inconsistencies that I have observed. CHEERS checklist: 1) Item 11a seems to be reported at page 7. 2) Item 15 seems to have been done in the article. 3) Item 18 is reported on page 12-13 from my understanding 4) Item 23 and 24 are reported on page 18 if I interpreted things correctly.
--	---

REVIEWER	Ella Zomer Monash University, Australia
REVIEW RETURNED	19-Jun-2017

GENERAL COMMENTS	This paper describes a cost and cost-effectiveness study of habit-based advice for weight control versus usual care in the Ten Top Tips (10TT) trial. This is an important study as the prevalence of obesity is continuing to rise. The study showed there were similar costs and QALYs in the 10TT and usual care arms. Overall this is a well conducted study. Specific comments are follows:  - For the 10TT arm, you added additional time that patients spent with the practice nurse to introduce the program. Were any additional time costs included for the usual care arm? Do you think that additional time (more than a standard consultation) is required when referring these patients to a standard weight loss intervention? As your results demonstrated higher NMB when usual care costs were added, adding these additional time costs could also increase your NMB. - In this study, your sensitivity analyses were focussed on the uncertainty around costs. Did you think about the uncertainty around other input variables such as the degree of weight loss and in turn QALYs. I'd be interested to see how your results would vary by including some sensitivity analyses around QALYs. - Due to the relatively small difference in weight loss over the trial follow-up period, as well as QALYs and resource use, I think the 'Discussion' section would benefit from discussion around
---

	contributors which may influence this, such as the healthy bias effect (particularly as it was unblinded), also the potential effect of GPs prescribing more services for those in the usual care arm than may typically be reported due to them being part of a trial. Also, the limitations of the EQ-5D when measuring differences in health.  - I think it would also benefit from discussion around why the complete case analysis results were so different. That is, the 10TT had much lower mean QALYs gained than the usual care group. - The table numbers are not in sequence in the main text. - In Table 1, sex is presented as a mean value with a confidence interval. From my understanding, these were differences in characteristics (such as age and sex) between groups at baseline so I would expect these to be presented as n (%). Also, could think about adding p-values to this table to show that the significance of differences between groups. - In Table 2, resource use and costs are presented for primary and secondary care services. It is unclear if this is for resource use before or after randomisation or both. - In Table 1 and 3, the key includes SD however SE is presented in the table. - Table 5 columns are not aligned correctly.
--	---

VERSION 1 – AUTHOR RESPONSE

Response to reviewers' comments:

Reviewer: 1

Reviewer Name: Dr Ed Wilson

Institution and Country: University of Cambridge, UK Competing Interests: None declared

General

Overall this appears to have been a well conducted and reported economic evaluation alongside a clinical trial.

My main concern is with the use of EQ5D based QALYs over a period of only two years in a weight-loss trial. Whilst there may be some short-term positive psychological effect of weight loss (eg Peckmezian & Hay J Eat Disord 2017 PMID: 28489914) which may be reflected in the EQ5D, the major impacts will be long term from avoidance of T2DM, arthritis and some cancers and their associated complications. Therefore, a 2-year time horizon will not capture these. No decision modelling was undertaken due to the lack of a significant treatment effect at 24 months (although in any case I note there were no plans to do so noted in the protocol). As per my comments below I think it is still important to report the mean difference and 95%CI as this would be useful for future meta-analysis. My preference would be to see some longer-term decision modelling based on the 24 month data, so that parameter uncertainty is translated into decision uncertainty, but I suspect the (probably small) added value to decision makers would not justify the effort. Perhaps the best way to handle this comment is to add a few sentences to the discussion.

- We did not undertake modelling beyond the end of the two-year follow-up as there was no impact on weight change at this point. We have stated in the discussion why there may be value in using other measures and how the time horizon may miss out on any potential future benefits from disease avoidance.

Specific

P4L75 – avoid imprecise, emotive terms such as ‘huge’: be specific. Perhaps simply delete the first sentence of the paragraph.

- This sentence has been deleted.

P4L76-77 “Health risks associated with obesity...” – this claim needs a reference (eg Guh et al. BMC Pub Health 2009 DoI: 10.1186/1471-2458-9-88)

- Referenced using the publication above (Guh et al, 2009).

P4L78-80 “as the prevalence...”: The NOO chart appears to show a levelling off of obesity prevalence since around 2008 – the rise is certainly a lot slower than between say 1994 and 2003.

- This sentence has been deleted.

P4L81 – P6L112: Comparison with other papers should be merged with the respective points in the discussion. A shorter concise introduction is better – perhaps just state that non-surgical interventions exist but there is little evidence on cost-effectiveness, and cite the papers but save the actual numbers for the discussion. A new paper of relevance to this is Ahern et al. Lancet 2017

PMID:28478041

- The Ahern et al (2017) reference above has been included in the introduction and publications from existing non-surgical interventions are mentioned in the discussion.

P6L126: Non-UK readers may not be familiar with the ‘five-a-day’ campaign. Suggest you provide a few words explaining.

- The five-a-day campaign is explained.

P7L134-137: It may be more informative to international readers to simply state the regions of England where recruitment took place and name the locations in the acknowledgements.

- Practice names and locations are replaced with regions. Exact locations can now be found in the acknowledgments section.

P7L144-145: please state the point estimate and 95%CI around the 24m outcome, rather than just that there is ‘no [significant] difference’.

- The point estimate has been inserted. The difference at 24 months was +0.75kg (CI: 95% (-0.73, 2.24)).

P8L158-160 ‘found no evidence of significant differences’: I am very wary of this line of reasoning – absence of proof is not proof of absence, and the definition (p-value cut-off) of what is statistically significant is in any case utterly arbitrary. Apart possibly from a boost to mental wellbeing, I would not expect the EQ5D to show any effect whatsoever from weight loss over 2 years. Likewise, with NHS costs – the major benefits will be in avoiding T2DM and its complications in 10+ years, so a 2 year time horizon will not capture these events. A preferable strategy would be to input the observed differences – and uncertainty - in weight loss at 2 years into a model and translate that parameter uncertainty through to decision uncertainty. The counter to this would be if there really was zero effect of the intervention by 2 years (proof of absence), and that the period of lower weight had no cumulative effect on future risk of obesity related disease. (This would be in contrast to T2DM, where I think there is some evidence that a period of tight blood glucose control in T2DM still has beneficial effects even if control subsequently deteriorates, presumably through a reduced accumulation of damage – from memory one of the UKPDS papers reported on this). Reporting the point estimate and 95%CI from the non-sig difference at 12 months would be helpful in justifying a decision not to model the longer term costs and outcomes.

- Given the findings of the clinical trial we think it would be inappropriate to model . Point estimate and 95% CI are now reported in the text and tables 1,2, 3 and 5. ‘Extrapolation beyond this time period was not undertaken because the within-trial analysis found no evidence of significant differences in

costs (-£36 (95%CI £511 to £440, p-value 0.88) or outcomes between groups (0 (95% CI -0.080 to 0.082, p-value 0.93) at 24 months...'

P8L164: typo: post or prior to randomisation? (same typo in Table 4?) Were any data extracted prior to randomisation (eg 3m prior to) as a 'baseline' cost to adjust for in regression analysis? (The later text suggests data were extracted 2yrs prior to 2 years post randomisation).

- Table 4 re-numbered to table 3. Data were extracted for the same participants two years prior to randomisation (before the trial started) and two years post randomisation (10TT trial). This has been reworded for clarity. Table 3 contains two years prior to randomisation and post randomisation. The 'prior to randomisation data' were extracted as a 'baseline' cost to adjust for in the regression analyses.

P8L166-167: Primary care records are notoriously bad at recording secondary care data. Why did you not access HES for these? (Too late to do anything about this now (!), but perhaps note this in the discussion).

- Hospital Episode Statistics (HES) data were suitable for 10TT. Given the short timeline for this trial it was not possible to link the patient-level data collected in the trial with HES data. Moreover, such linked data are often inaccurate (.Completeness and diagnostic validity of recording acute myocardial infarction events in primary care, hospital care, disease registry, and national mortality records: cohort study (BMJ 2013;346:f2350).

P10: Missing data: Please state the proportion of missing data for each parameter.

- The percentages of missing data for each variable are now reported in the manuscript.

P11L236: Is reference 21 correct? This appears to be a CUA, not a methods paper on INB. A better reference would be Zethraeus et al. Pharmacoeconomics 2003. DoI: 10.2165/00019053-200321010-00003

- Reference added.

P11L237-242: Utility, and therefore probably QALYs too, are usually left skewed, especially in a mostly healthy population. Would a GLM have provided a better fit to the data than OLS in this case?

- My understanding is that OLS minimises residual error and finds the best fit for the utilities and QALY data. The GLM would've resulted in the loss of precision and are reported to often be heavy left tailed. Manca et al (2005) suggest that OLS or GLM can be used in regression analysis for QALYs 'The more appropriate method of dealing with imbalance in mean baseline utilities is the use of multiple regression. This approach allows the estimation of differential QALYs, as well as the prediction of adjusted QALYs, while controlling for baseline utility values..... The regression-based approach not only generates an unbiased estimate of differential QALYs between the arms of the trial, but also increases the precision of the treatment effect estimate.....'

(<http://onlinelibrary.wiley.com/doi/10.1002/hec.944/epdf>)

Tables 2-4: I would strongly advise dropping the p-values from Tables 2-4. It would be better to report 95%CIs around differences, even better to limit increments to just total cost, or by sector too.

- Tables have been re-numbered. P-values have been dropped from tables 1, 3 and 5. P- values have been inserted in table 2 Difference between Costs and QALYs are reported with CI in tables 1,3 and 5 : 95%.

Minor:

P6L125: ?typo 'healthy' or 'health' snack?

- Corrected to healthy.

Throughout: Data are plural, not singular.

- This has been accounted for throughout

Reviewer: 2

Reviewer Name: Fredrik Norström

Institution and Country: Umeå University, Sweden Competing Interests: None declared

The article is very well written and addresses an important topic. Interestingly the study does not come up with the results that might have been expected from the authors, i.e. that the weight reduction intervention clearly is cost-effective. The result of the cost-effectiveness analysis, which to some extent could be called "negative" results should be important contribution to the field. The result indicates that only giving cost advices, at least in a similar way as described in the article, is not sufficient to have an effective weight reduction in the long run.

All of the revisions I request below are of rather minor character.

Methods:

1) Row 162: According to the CHEERS (item 9) statement should the discount rate be backed up by reference or similar.

- A reference from the National Institute of Health and Care Excellence (NICE) has been added.

2) Row 171-173: Missing end parenthesis.

- Missing bracket added.

3) Row 202: Specify that you use EuroQol 5D on first occasion instead of starting with the short form of the instrument.

- Changed to 'Health utilities were based on the EuroQol standardised measure of health status- EQ-5D-3L descriptive system'.

4) Row 215: "used" should likely be "was used".

- Changed to 'was used'.

5) Row 225-226: A reference is required for the use of "A linear regression model for imputed data" as this is a non-standard method.

- Two new references inserted- Sinharay et al (2001) and Yang et al (2005).

6) Row 227-228: It is unclear how and why non-parametric bootstrap has been used to calculate the difference in mean costs. It seems like the purpose with the bootstrap technique is to estimate the uncertainty (regardless if confidence intervals or graphical illustrations are the target for this) and not the mean costs. If so, revise the text accordingly. Bootstrap is not a well-known technique for any researcher so I recommend to add a reference for your method.

- New reference added in- Pasta et al (1999). A short description of the bootstrap technique to estimate uncertainty has been added.

7) Row 237-238: The description needs to be improved.

- Description improved with a sentence on what OLS is and the benefits of using OLS.

8) Row 247-248: I assume that the CIs are calculated based on normality as you use the standard errors but this is not specified. As you use Bootstrap estimates elsewhere would this information be valuable to make it clear.

- This has been updated. It should now be clear that CI were based on a normal distribution.

9) It is mentioned that you use CEAC but not cost-effectiveness planes in the method. Add such

information.

- A sentence on what the cost-effectiveness plane has been inserted- 'The cost-effectiveness plane were used to illustrate the difference in costs and outcomes between the two arms on a graph'.

Tables:

1) Table 1: CI should be explained with confidence interval in the footnotes as should SE be. The general recommendation is to present SD instead of SE as it is not sensitive to the sample size, thus enables numerical comparisons of potential differences in variation between results so your manuscript might benefit from such change.

- On the recommendation of our trial statistician we now report SDs when presenting non-imputed data and SEs when presenting imputed data. Our view is that it is better to present SEs for imputed data as these capture "how confident" you are about your estimated mean. We have reported SDs for non-imputed data and SEs for imputed data.

2) Table 2: SD = standard deviation is mentioned despite SE used in the table.

- As above. Non-imputed data changed to SD= standard deviation. Note: all tables have been re-numbered.

3) Table 4: The title is too unspecific in relation to the content of the table.

- Re-numbered to table 3 and re-titled to 'Table 3. Cost description two years prior and post randomisation'.

4) Table 5: It is not clear when imputed values have been used and not for the analyses in the table.

- The base case analysis and all subsequent analyses, except the complete case analysis in table 5 are based on imputed values.

References:

1) Reference 7 is a World Bank document, while the reference refer to results from a weight loss program. This needs to be corrected.

- This reference refers to the converted international costs and has now been moved to the discussion section.

2) Reference 20 does not fit on row 206.

- This reference has been deleted.

3) Reference 19 and 20 are identical. Please correct this.

- The duplicate reference has been deleted.

4) Check if both references 21 and 22 are valid as references for how to use multiple imputation which they both are indicating now.

- Both references are valid. They refer to a trial (GNOME) which undertakes multiple imputation and explains how to deal with imputed data (Briggs et al).

5) Row 223: References need to be fixed. Journal also likely to require city, state and country for the software here.

- New reference added- StataCorp. 2015. Stata Statistical Software: Release 14. College Station, TX: StataCorp LP.

6) Author name (P R) for reference 23 is not complete.

New reference added- Royston P. Multiple imputation of missing values: update of ice Stata J 2005: 527-36.

7) Check also other references to see if they are correctly specified as there are so many

inconsistencies that I have observed.

- All references have been updated.

CHEERS checklist:

- 1) Item 11a seems to be reported at page 7.
- 2) Item 15 seems to have been done in the article.
- 3) Item 18 is reported on page 12-13 from my understanding
- 4) Item 23 and 24 are reported on page 18 if I interpreted things correctly.

- The CHEERS checklist has been updated. The items from the checklist will not follow the suggestions above due to the track changes.

Reviewer: 3

Reviewer Name: Ella Zomer

Institution and Country: Monash University, Australia Competing Interests: None declared

This paper describes a cost and cost-effectiveness study of habit-based advice for weight control versus usual care in the Ten Top Tips (10TT) trial. This is an important study as the prevalence of obesity is continuing to rise. The study showed there were similar costs and QALYs in the 10TT and usual care arms.

Overall this is a well conducted study. Specific comments are follows:

- For the 10TT arm, you added additional time that patients spent with the practice nurse to introduce the program. Were any additional time costs included for the usual care arm? Do you think that additional time (more than a standard consultation) is required when referring these patients to a standard weight loss intervention? As your results demonstrated higher NMB when usual care costs were added, adding these additional time costs could also increase your NMB.

- No additional intervention costs were added to the usual arm because we did not record the amount of time spent referring patients (and this is likely to have varied). However a recent study suggests such a referral can be done briefly (within 30seconds):

[http://www.thelancet.com/journals/lancet/article/PIIS0140-6736\(16\)31893-1/fulltext](http://www.thelancet.com/journals/lancet/article/PIIS0140-6736(16)31893-1/fulltext)

- In this study, your sensitivity analyses were focussed on the uncertainty around costs. Did you think about the uncertainty around other input variables such as the degree of weight loss and in turn QALYs. I'd be interested to see how your results would vary by including some sensitivity analyses around QALYs.

- We have included a sensitivity analysis based on complete case data, but other than this, since the QALYs were based on the EQ-5D data collected directly in the trial it is difficult to think of a meaningful sensitivity analysis to run.

- Due to the relatively small difference in weight loss over the trial follow-up period, as well as QALYs and resource use, I think the 'Discussion' section would benefit from discussion around contributors which may influence this, such as the healthy bias effect (particularly as it was unblinded), also the potential effect of GPs prescribing more services for those in the usual care arm than may typically be reported due to them being part of a trial. Also, the limitations of the EQ-5D when measuring differences in health.

- The discussion states how over prescribing of standard weight-loss interventions in the usual care arm could affect the results. Furthermore, as we did not collect data on the amount of referrals that took place in usual care by GPs it is difficult to approximate how many over 'over referrals' there were. However, we undertook two analyses: (1) usual care group receiving additional care - referrals to

weight watcher/slimming world, down size programme from their GP (2) usual care group receiving no additional care. Both analyses show small differences in costs and QALYs.

We have stated the value in using other HRQoL measures instead of stating the disadvantages of the EQ-5D in the discussion.

- I think it would also benefit from discussion around why the complete case analysis results were so different. That is, the 10TT had much lower mean QALYs gained than the usual care group.
- The complete case analysis included participants for whom we had both costs and EQ-5D scores available for. The 10TT arm showed a small difference in the QALYs gained (-0.047 95% CI: -0.180 to 0.086 p-value=0.46), which was non-significant.

- The table numbers are not in sequence in the main text.
- Table sequence in the main text has been updated.

- In Table 1, sex is presented as a mean value with a confidence interval. From my understanding, these were differences in characteristics (such as age and sex) between groups at baseline so I would expect these to be presented as n (%). Also, could think about adding p-values to this table to show that the significance of differences between groups.
- Table 1 is now table 2 describes the characteristic of the participant and not results from the OLS or GLM for adjusted factors. We have removed mention of table one to avoid confusion. We have added p- values to indicate statistical significant differences between the two groups.

- In Table 2, resource use and costs are presented for primary and secondary care services. It is unclear if this is for resource use before or after randomisation or both.
- Table 2 is now table 1. The title for table 1 has been reworded to 'Resource use, unit cost, and mean cost per participant for primary and secondary care services post randomisation'.
- In Table 1 and 3, the key includes SD however SE is presented in the table.
- Note: the table sequence has changed. All footnotes with SD or SE contain the following: SD = standard deviation for non-imputed data. SE = standard error for imputed data to capture how confident we are about the mean value. CI = confidence interval
- Table 5 columns are not aligned correctly.
- Table 1-5 have been re-aligned.

VERSION 2 – REVIEW

REVIEWER	Fredrik Norström Umeå University, Sweden
REVIEW RETURNED	21-Aug-2017

GENERAL COMMENTS	General comments I still think that the value of the manuscript is obvious and that it is a well-designed study. However, after the revision it has become apparent that missing data is a major concern for the analyses in the manuscript and it therefore needs to be major improvement in how this is dealt with in the manuscript so that it possible to judge on the reliability of the results. Comments on revisions are listed below for both my own (reviewer 2) previous comments and the comments from the other reviewers.
---

	I have some additional comments as well that I want to start with. There are some important issues that needs to be handled and some of these are new since previous version as the new text has made these apparent. The most important is to well motivate how it can be ok to use variables with so high level of missing. This weakness must be well motivated if analyses based on imputations, which the manuscript heavily relies on, should be possible to use. If not it might be difficult to motivate the manuscript being published. The language for the revised part of the text needs to be improved. The language in original version was good but now there are some grammar errors that needs to be corrected at places. I have not listed these and expect the authors to go through the paper themselves. The method section is more difficult to follow after the revision and this needs to be checked. It is related to the language (see previous paragraph) to some extent. On row 118 there is inconsistency with “.” followed by “;”. From the text at 118-120 it is not obvious how 10TT is related to usual care after 24 months, which is the time period used in the manuscript. Row 132 is end parenthesis missing. At row 172-173 it is stated: “The unit costs for each cost component are shown in Table 1.”. What do you mean with “unit cost”? Be clearer. In results are Table 3 mentioned twice in the paragraph from row 253 to 257. Once is sufficient. I am not sure to move text in background to discussion. I don't mind that you bring up what is previously known. It would have been logical to lower the detail level of it though compared to original version. However, I think that suggestion from reviewer 1 is ok. Regardless, the text in the discussion must be used to relate your study to previous studies. In the new text it is not possible to understand how previous studies relates to yours. Even if the text is pretty extensive in size this must not necessarily be revised, but it must be apparent why this information is relevant in relation to your findings. This is something that has to be handled. At row 363, what do you mean with benefits? I think that you refer to utilities/QALYs.
--	--

VERSION 2 – AUTHOR RESPONSE

Editorial Comments:

We appreciate that reviewer 2's comments are quite negative; however, in light of the positive comments received from the other reviewers, we felt we should give you the opportunity to respond to the criticisms and revise your manuscript appropriately. Please note that we may ask the reviewer to assess your revised manuscript so urge you to address all comments as thoroughly as possible.

In addition to reviewer 2's comments we also received some feedback from reviewer 1 (included below), who recommended publication. We would be grateful if you could consider his remaining concerns in the next revision (we consider these to be optional revisions).

Comments from Reviewer 1:

The authors have done a good job responding to my comments with two minor points:

1. GLM vs OLS: I'm still of the opinion that GLM is preferable, but happy to defer to a statistician on this: I would agree that OLS provides the more precise estimate (lower SE), but that is not much use if it is precisely wrong! Given this particular situation though, I suspect any differences will be trivial and will not affect the results of the study at all.

Response: With such large sample, the sampling distribution of the mean difference (in cost and in QALY) should be normally distributed, and the OLS therefore valid (<http://onlinelibrary.wiley.com/doi/10.1002/hec.1653/full>). A GLM should be fine as well, but we preferred OLS for a few reasons of simplicity for interpreting a mean difference.

We've also undertaken the cost analysis using GLM and the results were significantly non-significant.

2. I'm not a fan of p-values alongside baseline values (Table 2). The CIs are probably unnecessary too. If px are randomised, any baseline differences must be by chance, irrespective of the long-run relative frequency with which that difference is likely to be observed under Ho. Although I guess if you have a cluster of low p-values in your baseline data it may suggest something fishy is going on.

Response: The p-values and CI have been removed from Table 2.

Reviewer Comments to Author:

Reviewer: 2

Reviewer Name: Fredrik Norström

Institution and Country: Umeå University, Sweden Competing Interests: None declared

Comments are available in attached file (see below).

General comments

I still think that the value of the manuscript is obvious and that it is a well-designed study. However, after the revision it has become apparent that missing data is a major concern for the analyses in the manuscript and it therefore needs to be major improvement in how this is dealt with in the manuscript so that it possible to judge on the reliability of the results.

Comments on revisions are listed below for both my own (reviewer 2) previous comments and the comments from the other reviewers.

I have some additional comments as well that I want to start with. There are some important issues that needs to be handled and some of these are new since previous version as the new text has made these apparent. The most important is to well motivate how it can be ok to use variables with so high level of missing. This weakness must be well motivated if analyses based on imputations, which the manuscript heavily relies on, should be possible to use. If not it might be difficult to motivate the manuscript being published.

Response: The benefits of using imputed data is discussed further in the discussion- see below.

"It is important to note that when using such methods there is uncertainty around the non-observed value across the imputations. To account for the uncertainty around the values, we employed the non-

parametric bootstrap approach to estimate the variance (a representation of uncertainty) around the true values (21). We acknowledge that though multiple imputation is able to statistically test for error, this method can produce bias. The bias arises from the assumption that missing data in the study were 'missing at random'. For example, the missing at random assumption may be reasonable if a variable that is predictive of missing data is included in the imputation model, but not if the variable is omitted from the model(35). However, multiple imputation offers an advantage over complete case analysis, by filling in missing values and enabling participants to be included in the analysis, where there were auxiliary variables that are strong predictors of missingness(36). Guidelines were followed to ensure the model predicted the best possible estimates; investigation of the proportion of missing data; selection of specific variables and; identification of the predictors of missingness. We acknowledge that as the percentage of missingness increases (in some cases more than 50%) there will be greater inefficiency in the imputed data and more chance of bias. However, in general the application of regression imputation with large missing data may be acceptable if adjustments are made for the predictions, as we have done(21)".

The language for the revised part of the text needs to be improved. The language in original version was good but now there are some grammar errors that needs to be corrected at places. I have not listed these and expect the authors to go through the paper themselves.

The method section is more difficult to follow after the revision and this needs to be checked. It is related to the language (see previous paragraph) to some extent. On row 118 there is inconsistency with "." followed by ":",

Response: ; has been removed.

From the text at 118- 120 it is not obvious how 10TT is related to usual care after 24 months, which is the time period used in the manuscript.

Response: This has been reworded to "At three months participants in the 10TT group lost significantly more weight than those receiving usual care (mean difference in weight change=-0.87kg, 95% CI -1.47 to -0.27, p=0.004). But this effect was not maintained at 24 months (mean difference in weight change= +0.75kg (95% CI: -0.73 to 2.24))."

Row 132 is end parenthesis missing.

Response: Parenthesis added.

At row 172-173 it is stated: "The unit costs for each cost component are shown in Table 1.". What do you mean with "unit cost"? Be clearer.

Response: Description of unit cost has been revised. It has been defined as "the total expenditure incurred by the NHS for one visit".

In results are Table 3 mentioned twice in the paragraph from row 253 to 257. Once is sufficient.

Response: Table 3 mentioned only once now in this paragraph.

I am not sure to move text in background to discussion. I don't mind that you bring up what is previously known. It would have been logical to lower the detail level of it though compared to original version. However, I think that suggestion from reviewer 1 is ok. Regardless, the text in the discussion must be used to relate your study to previous studies. In the new text it is not possible to understand

how previous studies relates to yours. Even if the text is pretty extensive in size this must not necessarily be revised, but it must be apparent why this information is relevant in relation to your findings. This is something that has to be handled

Response: The discussion has been revised to link previous studies to 10TT and what weight loss programs add to this study. Please refer to the discussion.

At row 363, what do you mean with benefits? I think that you refer to utilities/QALYs.

Response: Benefits has been changed to QALYs for consistency with rest of the paper.

Author's Response to Decision Letter (bmjopen-2017-017511)

Cost-effectiveness of habit-based advice for weight control versus usual care in general practice in the Ten Top Tips (10TT) Trial: economic evaluation based on a randomised controlled trial

Author's Response

Response to reviewers' comments:

Reviewer: 1

Reviewer Name: Dr Ed Wilson

Institution and Country: University of Cambridge, UK Competing Interests: None declared

General

Overall this appears to have been a well conducted and reported economic evaluation alongside a clinical trial.

My main concern is with the use of EQ5D based QALYs over a period of only two years in a weight-loss trial. Whilst there may be some short-term positive psychological effect of weight loss (eg Peckmezian & Hay J Eat Disord 2017 PMID: 28489914) which may be reflected in the EQ5D, the major impacts will be long term from avoidance of T2DM, arthritis and some cancers and their associated complications. Therefore, a 2-year time horizon will not capture these. No decision modelling was undertaken due to the lack of a significant treatment effect at 24 months (although in any case I note there were no plans to do so noted in the protocol). As per my comments below I think it is still important to report the mean difference and 95%CI as this would be useful for future meta-analysis. My preference would be to see some longer-term decision modelling based on the 24 month data, so that parameter uncertainty is translated into decision uncertainty, but I suspect the (probably small) added value to decision makers would not justify the effort. Perhaps the best way to handle this comment is to add a few sentences to the discussion.

- We did not undertake modelling beyond the end of the two-year follow-up as there was no impact on weight change at this point. We have stated in the discussion why there may be value in using other measures and how the time horizon may miss out on any potential future benefits from disease avoidance.
- REPLY: From the statement in discussion, it is difficult to understand why even a two-year time period is relevant. You kind of dismiss that even the two years period were useful with such statement.

Response: The discussion section on the limitation of using a two-year time period has been revised. “However, at the time of this study little was known about the association between habit-formation and weight loss. This study has identified the need for longer-term strategies for continued adherence of weight loss. With add-on approaches such as counselling and education, alongside 10TT it may be possible to maintain weight loss post two-years.”

Specific

P4L75 – avoid imprecise, emotive terms such as ‘huge’: be specific. Perhaps simply delete the first sentence of the paragraph.

- This sentence has been deleted.
- REPLY: Ok

P4L76-77 “Health risks associated with obesity...” – this claim needs a reference (eg Guh et al. BMC Pub Health 2009 DoI: 10.1186/1471-2458-9-88)

- Referenced using the publication above (Guh et al, 2009).
- REPLY: Ok

P4L78-80 “as the prevalence...”: The NOO chart appears to show a levelling off of obesity prevalence since around 2008 – the rise is certainly a lot slower than between say 1994 and 2003.

- This sentence has been deleted.
- REPLY: Ok. Delete “.” from that sentence as well.

Response: “.” has been deleted

P4L81 – P6L112: Comparison with other papers should be merged with the respective points in the discussion. A shorter concise introduction is better – perhaps just state that non-surgical interventions exist but there is little evidence on cost-effectiveness, and cite the papers but save the actual numbers for the discussion. A new paper of relevance to this is Ahern et al. Lancet 2017 PMID:28478041

- The Ahern et al (2017) reference above has been included in the introduction and publications from existing non-surgical interventions are mentioned in the discussion.
- REPLY: Ok. However, limited should be limited.

P6L126: Non-UK readers may not be familiar with the ‘five-a-day’ campaign. Suggest you provide a few words explaining.

- The five-a-day campaign is explained.
- REPLY: Ok

P7L134-137: It may be more informative to international readers to simply state the regions of England where recruitment took place and name the locations in the acknowledgements.

- Practice names and locations are replaced with regions. Exact locations can now be found in the acknowledgments section.
- REPLY: Ok.

P7L144-145: please state the point estimate and 95%CI around the 24m outcome, rather than just that there is ‘no [significant] difference’.

- The point estimate has been inserted. The difference at 24 months was +0.75kg (CI: 95% (-0.73, 2.24)).
- REPLY: Ok.

P8L158-160 'found no evidence of significant differences': I am very wary of this line of reasoning – absence of proof is not proof of absence, and the definition (p-value cut-off) of what is statistically significant is in any case utterly arbitrary. Apart possibly from a boost to mental wellbeing, I would not expect the EQ5D to show any effect whatsoever from weight loss over 2 years. Likewise, with NHS costs – the major benefits will be in avoiding T2DM and its complications in 10+ years, so a 2 year time horizon will not capture these events. A preferable strategy would be to input the observed differences – and uncertainty - in weight loss at 2 years into a model and translate that parameter uncertainty through to decision uncertainty. The counter to this would be if there really was zero effect of the intervention by 2 years (proof of absence), and that the period of lower weight had no cumulative effect on future risk of obesity related disease. (This would be in contrast to T2DM, where I think there is some evidence that a period of tight blood glucose control in T2DM still has beneficial effects even if control subsequently deteriorates, presumably through a reduced accumulation of damage – from memory one of the UKPDS papers reported on this). Reporting the point estimate and 95%CI from the non-sig difference at 12 months would be helpful in justifying a decision not to model the longer term costs and outcomes.

- Given the findings of the clinical trial we think it would be inappropriate to model. Point estimate and 95% CI are now reported in the text and tables 1,2, 3 and 5. 'Extrapolation beyond this time period was not undertaken because the within-trial analysis found no evidence of significant differences in costs (-£36 (95% CI £511 to £440, p-value 0.88) or outcomes between groups (0 (95% CI -0.080 to 0.082, p-value 0.93) at 24 months...'
- REPLY: Ok, but correct 0 to 0.000 (or whatever the estimate is with three decimals exactness) as it is difficult to read without better precision.

Response: Changed to 0.000 for precision.

P8L164: typo: post or prior to randomisation? (same typo in Table 4?) Were any data extracted prior to randomisation (eg 3m prior to) as a 'baseline' cost to adjust for in regression analysis? (The later text suggests data were extracted 2yrs prior to 2 years post randomisation).

- Table 4 re-numbered to table 3. Data were extracted for the same participants two years prior to randomisation (before the trial started) and two years post randomisation (10TT trial). This has been reworded for clarity. Table 3 contains two years prior to randomisation and post randomisation. The 'prior to randomisation data' were extracted as a 'baseline' cost to adjust for in the regression analyses.
- REPLY: Ok.

P8L166-167: Primary care records are notoriously bad at recording secondary care data. Why did you not access HES for these? (Too late to do anything about this now (!), but perhaps note this in the discussion).

- Hospital Episode Statistics (HES) data were suitable for 10TT. Given the short timeline for this trial it was not possible to link the patient-level data collected in the trial with HES data. Moreover, such linked data are often inaccurate. Completeness and diagnostic validity of recording acute myocardial infarction events in primary care, hospital care, disease registry, and national mortality records: cohort study (BMJ 2013;346:f2350).

- REPLY: Ok.

P10: Missing data: Please state the proportion of missing data for each parameter.

- The percentages of missing data for each variable are now reported in the manuscript.
- REPLY: Inserting missing data was very important considering the high level of missing data. As there is such high level of missing and it is solved with multiple imputation it needs to be clarified from the authors in the discussion how they can motivate to use variables with over 50% missing data, and in fact also more than 25% which is the case for most variables. How do they motivate that this is reasonable, it is a bigger weakness than is obvious in the discussion.

Response: The benefits of using imputed data is discussed further in the discussion- see below.

“It is important to note that when using such methods there is uncertainty around the non-observed value across the imputations. To account for the uncertainty around the values, we employed the non-parametric bootstrap approach to estimate the variance (a representation of uncertainty) around the true values (21). We acknowledge that though multiple imputation is able to statistically test for error, this method can produce bias. The bias arises from the assumption that missing data in the study were ‘missing at random’. For example, the missing at random assumption may be reasonable if a variable that is predictive of missing data is included in the imputation model, but not if the variable is omitted from the model(35). However, multiple imputation offers an advantage over complete case analysis, by filling in missing values and enabling participants to be included in the analysis, where there were auxiliary variables that are strong predictors of missingness(36). Guidelines were followed to ensure the model predicted the best possible estimates; investigation of the proportion of missing data; selection of specific variables and; identification of the predictors of missingness. We acknowledge that as the percentage of missingness increases (in some cases more than 50%) there will be greater inefficiency in the imputed data and more chance of bias. However, in general the application of regression imputation with large missing data may be acceptable if adjustments are made for the predictions, as we have done(21)”.

P11L236: Is reference 21 correct? This appears to be a CUA, not a methods paper on INB. A better reference would be Zethraeus et al. *Pharmacoeconomics* 2003. DoI: 10.2165/00019053-200321010-00003

- Reference added.
- REPLY: Ok.

P11L237-242: Utility, and therefore probably QALYs too, are usually left skewed, especially in a mostly healthy population. Would a GLM have provided a better fit to the data than OLS in this case?

- My understanding is that OLS minimises residual error and finds the best fit for the utilities and QALY data. The GLM would’ve resulted in the loss of precision and are reported to often be heavy left tailed. Manca et al (2005) suggest that OLS or GLM can be used in regression analysis for QALYs ‘The more appropriate method of dealing with imbalance in mean baseline utilities is the use of multiple regression. This approach allows the estimation of differential QALYs, as well as the prediction of adjusted QALYs, while controlling for baseline utility values..... The regression- based approach not only generates an unbiased estimate of differential QALYs between the arms of the trial, but also increases the precision of the treatment effect estimate.....’
(<http://onlinelibrary.wiley.com/doi/10.1002/hec.944/epdf>)

- REPLY: Ok.

Tables 2-4: I would strongly advise dropping the p-values from Tables 2-4. It would be better to report

95% CIs around differences, even better to limit increments to just total cost, or by sector too.

- Tables have been re-numbered. P-values have been dropped from tables 1, 3 and 5. p-values have been inserted in table 2. Difference between Costs and QALYs are reported with CI in tables 1, 3 and 5 : 95%.
- REPLY: Ok.

Minor:

P6L125: ?typo 'healthy' or 'health' snack?

- Corrected to healthy.
- REPLY: Ok

Throughout: Data are plural, not singular.

- This has been accounted for throughout
- REPLY: I have not assessed this specifically in my review.

Reviewer: 2

Reviewer Name: Fredrik Norström

Institution and Country: Umeå University, Sweden Competing Interests: None declared

The article is very well written and addresses an important topic. Interestingly the study does not come up with the results that might have been expected from the authors, i.e. that the weight reduction intervention clearly is cost-effective. The result of the cost-effectiveness analysis, which to some extent could be called "negative" results should be important contribution to the field. The result indicates that only giving cost advices, at least in a similar way as described in the article, is not sufficient to have an effective weight reduction in the long run.

All of the revisions I request below are of rather minor character.

Methods:

1) Row 162: According to the CHEERS (item 9) statement should the discount rate be backed up by reference or similar.

A reference from the National Institute of Health and Care Excellence (NICE) has been added.

- REPLY: Ok.

2) Row 171-173: Missing end parenthesis.

Missing bracket added.

- REPLY: Ok.

3) Row 202: Specify that you use EuroQol 5D on first occasion instead of starting with the short form of the instrument.

Changed to 'Health utilities were based on the EuroQol standardised measure of health status- EQ-5D-3L descriptive system'.

- REPLY: Ok

4) Row 215: "used" should likely be "was used".

Changed to 'was used'.

• REPLY: Ok.

5) Row 225-226: A reference is required for the use of "A linear regression model for imputed data" as this is a non-standard method.

Two new references inserted- Sinharay et al (2001) and Yang et al (2005).

• REPLY: Ok.

6) Row 227-228: It is unclear how and why non-parametric bootstrap has been used to calculate the difference in mean costs. It seems like the purpose with the bootstrap technique is to estimate the uncertainty (regardless if confidence intervals or graphical illustrations are the target for this) and not the mean costs. If so, revise the text accordingly. Bootstrap is not a well-known technique for any researcher so I recommend to add a reference for your method.

New reference added in- Pasta et al (1999). A short description of the bootstrap technique to estimate uncertainty has been added.

• REPLY: Ok.

7) Row 237-238: The description needs to be improved.

Description improved with a sentence on what OLS is and the benefits of using OLS.

REPLY: "which assumes a normal distribution and minimizes residual error" is not needed as this detailed explanation on why a linear regression is used is not standard, still not wrong to be this detailed though. The last part of the sentence is not correct and needs to be revised. You are not measuring the difference in values between QALY and dependent variables, which is stated now. I assume that what you do is to measure difference in QALY for 10TT and usual care.

Response: This has been corrected- "The ordinary least squares (OLS) method, which assumes a normal distribution was applied to estimate the mean difference in observed QALYs between two arms using regression analysis".

8) Row 247-248: I assume that the CIs are calculated based on normality as you use the standard errors but this is not specified. As you use Bootstrap estimates elsewhere would this information be valuable to make it clear.

This has been updated. It should now be clear that CI were based on a normal distribution.

• REPLY: Ok.

9) It is mentioned that you use CEAC but not cost-effectiveness planes in the method. Add such information.

A sentence on what the cost-effectiveness plane has been inserted- 'The cost- effectiveness plane were used to illustrate the difference in costs and outcomes between the two arms on a graph'.

• REPLY: Ok.

Tables:

1) Table 1: CI should be explained with confidence interval in the footnotes as should SE be. The general recommendation is to present SD instead of SE as it is not sensitive to the sample size, thus enables numerical comparisons of potential differences in variation between results so your

manuscript might benefit from such change.

On the recommendation of our trial statistician we now report SDs when presenting non-imputed data and SEs when presenting imputed data. Our view is that it is better to present SEs for imputed data as these capture “how confident” you are about your estimated mean. We have reported SDs for non-imputed data and SEs for imputed data.

- REPLY: I am not convinced that SEs is recommendable for imputed data but I consider it ok to use. Delete the explanation for why SE is used in the table. Such explanation can be given in methods section if you consider it as important. I think that it is not important enough to be mentioned there either. Be consistent with number of value digits ($p=0.5$ and $p=0.41$) in this table and throughout the article. There is too big precision given for confidence intervals, either use 57 (my suggestion) or 56.8.

Response: Explanation of SE removed from tables 1,3 and 5. The p-values from the table have been adjusted to the removed. The CI have been rounded up, where applicable.

2) Table 2: SD = standard deviation is mentioned despite SE used in the table.

As above. Non-imputed data changed to SD= standard deviation. Note: all tables have been re-numbered.

- REPLY: Ok.

3) Table 4: The title is too unspecific in relation to the content of the table.

Re-numbered to table 3 and re-titled to ‘Table 3. Cost description two years prior and post randomisation’.

- REPLY: New title ok. However, title of Table 2 needs to be improved. It now says that it presents differences between groups which it is not even doing (it can be calculated by hand from information in table though). Could you write “Demographics at baseline” instead of current title?

Response: The title for Table 2 has been changed to Demographics at baseline.

4) Table 5: It is not clear when imputed values have been used and not for the analyses in the table.

The base case analysis and all subsequent analyses, except the complete case analysis in table 5 are based on imputed values.

- REPLY: There are too detailed information about method-related issues in the table. Explanations should be kept for method section in manuscript. In both new Table 4 and Table 5 are the specification for SE too detailed (see previous comment). The content in Table 5 is unclear. What is measure at rows with “3 months”, “6 months” and so on. Isn’t it QALYs? I assume that discounted QALYs relates to aggregated QALYs for two years and this needs to be clarified as a QALY score is maximum 1 and average therefore cannot be above 1. The title mentions both utilities and QALYs. QALYs is a utility so current title is indirectly specifying QALYs twice and needs to be rewritten.

Response: Tables 4 and 5 no longer state why there is use of standard errors for imputed data. The remaining text below table 5 has been kept to aid readers.

From 3-24 months we’re measuring the utility score. The utility score cannot be above 1. 1= best possible health and 0= death. Combining the utility score across these time points (3-24 months) gives us the aggregate QALY (1 QALY= one year of improved quality and quality of life). For clarity, I’ve stated what the months represent (utility).

Table 5 has been reworded to “Quality Adjusted Life Years (QALYs) per patient to highlight that the focus of this table is QALYs.

References:

1) Reference 7 is a World Bank document, while the reference refer to results from a weight loss program. This needs to be corrected.

This reference refers to the converted international costs and has now been moved to the discussion section.

• REPLY: Ok

2) Reference 20 does not fit on row 206.

This reference has been deleted.

• REPLY: Ok.

3) Reference 19 and 20 are identical. Please correct this.

The duplicate reference has been deleted.

• REPLY: Ok.

4) Check if both references 21 and 22 are valid as references for how to use multiple imputation which they both are indicating now.

Both references are valid. They refer to a trial (GNOME) which undertakes multiple imputation and explains how to deal with imputed data (Briggs et al).

• REPLY: Ok.

5) Row 223: References need to be fixed. Journal also likely to require city, state and country for the software here.

New reference added- StataCorp. 2015. Stata Statistical Software: Release 14. College Station, TX: StataCorp LP.

• REPLY: The standard is not to add a reference for the use of software but only to write “Stata SE version 14 (Stata Corp, College Station, TX)” and not “Stata SE version 14(24)”. See other publications for guidance on how these are referred to.

Response: This has been updated to “Imputations were undertaken using the `–mi impute mvn–` command in Stata SE version 14 (Stata Corp, College Station, TX)”.
I have kept the reference for STATA in the references as I found that some papers published by BMJ Open do include software in the reference section.

6) Author name (P R) for reference 23 is not complete.

New reference added- Royston P. Multiple imputation of missing values: update of ice Stata J 2005: 527-36.

• REPLY: Ok.

7) Check also other references to see if they are correctly specified as there are so many inconsistencies that I have observed.

All references have been updated.

- REPLY: I haven't noted any inconsistencies so it seems to have been solved.

CHEERS checklist:

Item 11a seems to be reported at page 7.

Item 15 seems to have been done in the article.

Item 18 is reported on page 12-13 from my understanding

Item 23 and 24 are reported on page 18 if I interpreted things correctly.

The CHEERS checklist has been updated. The items from the checklist will not follow the suggestions above due to the track changes.

- REPLY: Ok.

Reviewer: 3

Reviewer Name: Ella Zomer

Institution and Country: Monash University, Australia Competing Interests: None declared

This paper describes a cost and cost-effectiveness study of habit-based advice for weight control versus usual care in the Ten Top Tips (10TT) trial. This is an important study as the prevalence of obesity is continuing to rise. The study showed there were similar costs and QALYs in the 10TT and usual care arms.

Overall this is a well conducted study. Specific comments are follows:

- For the 10TT arm, you added additional time that patients spent with the practice nurse to introduce the program. Were any additional time costs included for the usual care arm? Do you think that additional time (more than a standard consultation) is required when referring these patients to a standard weight loss intervention? As your results demonstrated higher NMB when usual care costs were added, adding these additional time costs could also increase your NMB.

- No additional intervention costs were added to the usual arm because we did not record the amount of time spent referring patients (and this is likely to have varied). However a recent study suggests such a referral can be done briefly (within 30seconds):

[http://www.thelancet.com/journals/lancet/article/PIIS0140-6736\(16\)31893-1/fulltext](http://www.thelancet.com/journals/lancet/article/PIIS0140-6736(16)31893-1/fulltext)

- REPLY: Ok.

- In this study, your sensitivity analyses were focussed on the uncertainty around costs. Did you think about the uncertainty around other input variables such as the degree of weight loss and in turn QALYs. I'd be interested to see how your results would vary by including some sensitivity analyses around QALYs.

- We have included a sensitivity analysis based on complete case data, but other than this, since the QALYs were based on the EQ-5D data collected directly in the trial it is difficult to think of a meaningful sensitivity analysis to run.

- REPLY: It is unproblematic to make a sensitivity analysis for QALYs. It is only to assume, e.g. a change in effect of 5% for 10TT. I am not sure though that such analysis would have an important added value and nothing that I request you to do.

- Due to the relatively small difference in weight loss over the trial follow-up period, as well as QALYs and resource use, I think the 'Discussion' section would benefit from discussion around contributors which may influence this, such as the healthy bias effect (particularly as it was unblinded), also the potential effect of GPs prescribing more services for those in the usual care arm than may typically be reported due to them being part of a trial. Also, the limitations of the EQ-5D when measuring

differences in health.

- The discussion states how over prescribing of standard weight-loss interventions in the usual care arm could affect the results. Furthermore, as we did not collect data on the amount of referrals that took place in usual care by GPs it is difficult to approximate how many over 'over referrals' there were. However, we undertook two analyses: (1) usual care group receiving additional care - referrals to weight watcher/slimming world, down size programme from their GP (2) usual care group receiving no additional care. Both analyses show small differences in costs and QALYs.
- We have stated the value in using other HRQoL measures instead of stating the disadvantages of the EQ-5D in the discussion.
- REPLY: Even if you lack the data, it is possibly to argue around these issues. I don't require you to do so though.

- I think it would also benefit from discussion around why the complete case analysis results were so different. That is, the 10TT had much lower mean QALYs gained than the usual care group.

- The complete case analysis included participants for whom we had both costs and EQ-5D scores available for. The 10TT arm showed a small difference in the QALYs gained (-0.047 95% CI: -0.180 to 0.086 p-value=0.46), which was non-significant.
- REPLY: I think that it is important to reason around this in the discussion as I already have mentioned you have MANY missing data for most variables. Imputing missing data might from the base case analysis indicate that these results are not trustable as the difference is large between the estimates. This needs to be better pinpointed.

Response: The benefits of using imputed data is discussed further in the discussion- see below.

"It is important to note that when using such methods there is uncertainty around the non-observed value across the imputations. To account for the uncertainty around the values, we employed the non-parametric bootstrap approach to estimate the variance (a representation of uncertainty) around the true values (21). We acknowledge that though multiple imputation is able to statistically test for error, this method can produce bias. The bias arises from the assumption that missing data in the study were 'missing at random'. For example, the missing at random assumption may be reasonable if a variable that is predictive of missing data is included in the imputation model, but not if the variable is omitted from the model(35). However, multiple imputation offers an advantage over complete case analysis, by filling in missing values and enabling participants to be included in the analysis, where there were auxiliary variables that are strong predictors of missingness(36). Guidelines were followed to ensure the model predicted the best possible estimates; investigation of the proportion of missing data; selection of specific variables and; identification of the predictors of missingness. We acknowledge that as the percentage of missingness increases (in some cases more than 50%) there will be greater inefficiency in the imputed data and more chance of bias. However, in general the application of regression imputation with large missing data may be acceptable if adjustments are made for the predictions, as we have done(21)".

- The table numbers are not in sequence in the main text.

- Table sequence in the main text has been updated.
- REPLY: Ok.

- In Table 1, sex is presented as a mean value with a confidence interval. From my understanding, these were differences in characteristics (such as age and sex) between groups at baseline so I would expect these to be presented as n (%). Also, could think about adding p-values to this table to

show that the significance of differences between groups.

- Table 1 is now table 2 describes the characteristic of the participant and not results from the OLS or GLM for adjusted factors. We have removed mention of table one to avoid confusion. We have added p- values to indicate statistical significant differences between the two groups.
- REPLY: See previous comments about this table.

Response: Based on recommendations from two statisticians and 1 reviewer, I've decided to exclude p- values form table 1 (now table 2).

An asterix detailing sex is presented as a proportion has been inserted below table 2 for clarity.

- In Table 2, resource use and costs are presented for primary and secondary care services. It is unclear if this is for resource use before or after randomisation or both.

• Table 2 is now table 1. The title for table 1 has been reworded to 'Resource use, unit cost, and mean cost per participant for primary and secondary care services post randomisation'.

• REPLY: Ok.

- In Table 1 and 3, the key includes SD however SE is presented in the table.

• Note: the table sequence has changed. All footnotes with SD or SE contain the following: SD = standard deviation for non-imputed data. SE = standard error for imputed data to capture how confident we are about the mean value. CI = confidence interval

• REPLY: See previous comment.

- Table 5 columns are not aligned correctly.

• Table 1-5 have been re-aligned

• REPLY: Ok.

VERSION 3 – REVIEW

REVIEWER	Fredrik Norström Umeå University, Sweden
REVIEW RETURNED	11-Dec-2017

GENERAL COMMENTS	New review of reviewer 2 Due to the responses to initial and re-review comments I have condensed my responses to a new list below. The responses to most previous comments have been well handled. The key part that needs to be improved is the comparisons with previous research. Current text only/mainly informs about what other studies have shown before but is not reflecting on how they compare with the results in the manuscript itself. It can also be argued if the informative text should rather be in introduction than discussion. Obvious is that the text is more detailed than is reasonable in relation to the study performed in the manuscript. General 1. There are still some linguistic errors in the text but it has been much improved. See e.g. row 223 where words are in wrong order,
--

grammar at row 217, no space before counterweight at row 336, at row 352 are things messed up between sentences and row 378 need to be rephrased (“are” twice in it). I also doubt that “Except” is correctly used in footnotes of Table 4. I strongly advice that a native language speaker read through the whole text and strengthens the text.

2. The reasoning around imputations of missing values in the discussion handles the limitations of the use of it well enough in revised version.

Introduction

I have no requests of improvements for the introduction section after the revisions made by the authors. However, I would have preferred to have a good motivation why the study is performed with information on what similar evaluations that have been performed previously for weight reduction based on similar programs. Though, I think this would be valuable the authors are not requested to make such change in their manuscript.

Methods

The method section is much easier to read in the revised version. I have a few minor revisions:

1. In trial background (starting at row 102), give a reference to the article where the methods and similar were first described, i.e. the RCT you use as basis for your study. In such reference the exact clinics and such, (that were specified in a previous version, should be accessible I assume. Nevertheless, it should be more informative than the information mentioned in your method section.
2. GLM vs OLS comment: Your explanation is ok for me. However, I recommend to improve the description so that it is clear for what independent variables the analysis is performed.
3. At row 195, 69% must be wrong. I guess it should be 31%. Also specify in first parenthesis or in text ahead of it that the % in parenthesis is specifying % missing to avoid that readers does not understand. With no specific item reported with more than 50% missing the weaknesses of the analyses is smaller than I previously assumed them to be.

Results

I have no requests besides the ones linked to comments below for tables. It is however a weakness that Table 3 compares values at baseline respectively follow-up between usual care and 10TT and not the change from before to after which would favor 10TT and perhaps even give significant results.

Discussion

1. You write “Our findings showed that weight loss programmes such as Weight Watchers lost more weight than education programmes and resulted in the possibility of cost savings, which we outline below”. This statement implies that results have been presented in regard to Weight Watchers in the manuscripts result section which is not the case. This sentence needs to be rewritten or there need to be something written about Weight Watchers (or similar) in results section. I suspect that the problem is that “Our

	findings” refer to what you find in previous publications and not to your own analyses. Then you need to rewrite the sentence so that this is clear. 2. Short form CP (after Weight Watchers) is not explained. Either use full form everywhere or explain what CP is. I suspect that CP=commercial product but I am not sure. 3. Row 298-321: The information about the results from the Weight Watchers study is very long. I suggest to shorten it a lot and only mention the key message in relation to 10TT. 4. A lot of information is provided about other studies but no information is given in comparison to the study performed in the manuscript. The section needs to be rewritten so that the relation between previous research and the submitted study is discussed. Now it is a lot of information not only in regard to comment 3 but also in regard to other previous studies. Acknowledgements The first of this section is not really written to acknowledge but rather to inform more about the study. The author’s would benefit from rewriting it accordingly. However, it can be kept as it is. Tables 1. N refers to the full population. You are using a sample of the population in your tables so therefore should n be used and the headers in tables be updated accordingly. 2. The reference to table 2 at row 218 is not in line with the content of it. A statement which is referring to the actual content of Table 2 is needed. 3. In Table 4, as well as in abstract and results, two value digits is required used. Especially important for the incremental utility as it is now stated as 0.000 while it must be slightly above 0 considering that the NMB is above 0. 4. In Table 5, it is still not clear that discounted QALY’s refer to two years, which is my interpretation. Cheers statement checklist Non-important adjustments: i) 17 should be 9-11, ii) 22 should be 14-19 I assume and iii) 23 and 24 are only on page 20 (assuming that “Competing interests” corresponds to “Conflicts of interest”).
--	--

VERSION 3 – AUTHOR RESPONSE

New review of reviewer 2

Due to the responses to initial and re-review comments I have condensed my responses to a new list below. The responses to most previous comments have been well handled. The key part that needs to be improved is the comparisons with previous research. Current text only/mainly informs about what other studies have shown before but is not reflecting on how they compare with the results in the manuscript itself. It can also be argued if the informative text should rather be in introduction than discussion. Obvious is that the text is more detailed than is reasonable in relation to the study performed in the manuscript.

General

1. There are still some linguistic errors in the text but it has been much improved. See e.g. row 223 where words are in wrong order, grammar at row 217, no space before counterweight at row 336, at row 352 are things messed up between sentences and row 378 need to be rephrased (“are” twice in it). I also doubt that “Except” is correctly used in footnotes of Table 4. I strongly advice that a native language speaker read through the whole text and strengthens the text.

- row 223 where words are in wrong order

Changed to “The bootstrapped results were combined using the formula described by Briggs et al (23), to calculate the mean values for costs and utilities and the standard errors around the imputed values.”

- grammar at row 217

Changed to “Incremental costs were initially adjusted for; age, gender, practice and costs two years prior to implementation of 10TT. Adjusting for these factors did not make a significant difference to the results. Therefore, these were not included in the final model”.

- no space before counterweight at row 336

Space added before Counterweight

- at row 352 are things messed up between sentences

This sentence has been deleted

- row 378 need to be rephrased (“are” twice in it)

Rephrased and changed to “Fourth, while there are various instruments available to measure health related quality of life, we administered the EQ-5D(31).”

- I also doubt that “Except” is correctly used in footnotes of Table 4

Except has been deleted from the footnotes

2. The reasoning around imputations of missing values in the discussion handles the limitations of the use of it well enough in revised version.

No further comment.

Introduction

I have no requests of improvements for the introduction section after the revisions made by the authors. However, I would have preferred to have a good motivation why the study is performed with information on what similar evaluations that have been performed previously for weight reduction based on similar programs. Though, I think this would be valuable the authors are not requested to make such change in their manuscript.

No further comment.

Methods

The method section is much easier to read in the revised version. I have a few minor revisions:

1. In trial background (starting at row 102), give a reference to the article where the methods and similar were first described, i.e. the RCT you use as basis for your study. In such reference the exact

clinics and such, (that were specified in a previous version, should be accessible I assume. Nevertheless, it should be more informative than the information mentioned in your method section.

Reference inserted and exact clinics stated- "The 10TT trial was a two-arm, individually randomised, controlled trial in which 537 obese men and women were enrolled(13). Practices across England (n=14) were recruited through the MRC General Practice Research Framework (GPRF). They were located in: Wellingborough, Southampton, Bradford-on-Avon, Bromsgrove, Frome, Guisborough, Glastonbury, Ivybridge, Dunstable, Liskeard, Ledbury, New Mills, and London."

2. GLM vs OLS comment: Your explanation is ok for me. However, I recommend to improve the description so that it is clear for what independent variables the analysis is performed.

The following independent variables were used to predict the incremental costs and QALYs; age, gender, practice, prior costs from the previous two years and baseline utility values. Adjusting for these variables did not make statistically significant differences. P values were calculated for the independent variables, these were non-significant.

"A linear regression model using ordinary least squares (OLS) regression was used in the final analyses to estimate the mean difference in observed QALYs between the two trial arms. The independent variables used to perform the analysis were age, gender, practice and baseline utility values. There were no differences between the two groups in terms of age, practice, gender, prior costs and baseline EQ-5D scores (P= 0.47, 0.65, 0.29 and 0.41) (Table 2). Therefore, these were not included in the final model."

3. At row 195, 69% must be wrong. I guess it should be 31%. Also specify in first parenthesis or in text ahead of it that the % in parenthesis is specifying % missing to avoid that readers does not understand. With no specific item reported with more than 50% missing the weaknesses of the analyses is smaller than I previously assumed them to be.

Outpatient service visits has been changed to 30% and a parenthesis has been added to specify I am stating the percentage of missing data.

This has been corrected. The text now reads "We imputed missing data (% of missing data) for the following variables: weight (0.2%); BMI (0.2%); waist circumference (0.6%); EQ-5D at baseline (5%), 3 months (26%), 6 months (40%), 12 months (46%) and 24 months (46%); NHS visits (GP practice (28%); GP home (31%); GP phone calls (30%); practice nurse (27%); nurse home (31%); nurse phone calls (31%); extra nurse (30%); dietician (31%); hospital inpatient stay (31%); outpatient clinic (30%); A&E visits (30%)) and other outpatient service visits (30%) (Table 1)."

Results

I have no requests besides the ones linked to comments below for tables. It is however a weakness that Table 3 compares values at baseline respectively follow-up between usual care and 10TT and not the change from before to after which would favor 10TT and perhaps even give significant results.

We acknowledge this was a limitation to the study and may have produced significant results in favour of 10TT. T-tests were carried out to detect significant differences between costs prior and post implementation of 10TT (Table 3), which were non-significant.

No further comment.

Discussion

1. You write “Our findings showed that weight loss programmes such as Weight Watchers lost more weight than education programmes and resulted in the possibility of cost savings, which we outline below”. This statement implies that results have been presented in regard to Weight Watchers in the manuscripts result section which is not the case. This sentence needs to be rewritten or there need to be something written about Weight Watchers (or similar) in results section. I suspect that the problem is that “Our findings” refer to what you find in previous publications and not to your own analyses. Then you need to rewrite the sentence so that this is clear.

This section has been rewritten as suggested- “We found that that commercial weight loss programmes were highly prescribed amongst primary care providers and these participants lost more weight than those in self-led education programmes alone”

2. Short form CP (after Weight Watchers) is not explained. Either use full form everywhere or explain what CP is. I suspect that CP=commercial product but I am not sure.

Short form CP (commercial product) has been deleted. It is replaced with Weight Watchers throughout the paper.

3. Row 298-321: The information about the results from the Weight Watchers study is very long. I suggest to shorten it a lot and only mention the key message in relation to 10TT.

The results from the Weight Watchers study have been shortened.

“We undertook a rapid review to compare our results to similar weight loss programs. We found that that commercial weight loss programmes were highly prescribed amongst primary care providers and these participants lost more weight than those in self-led education programmes alone. Fuller et al(7) reported the long-term analysis of a 20 minute GP consultation versus Weight Watchers and found Weight Watchers produced a cost saving of US\$47 per patient and an incremental 0.03 QALY gained per patient(6). Similarly, a recent evaluation in the UK of a primary care led behavioural programme(29) looked at a brief advice and self-help materials (primary care led programme) versus Weight Watchers over 12 weeks and over 52 weeks. The authors concluded that Weight Watchers was more effective over 12 weeks (-4.75kg) and 52 weeks (-6.76kg) than brief advice and self-help material (-3.26kg), at a cost of £159 per kg lost. Additionally, a primary care led program Counterweight (a nurse-delivered patient education programme) showed that nurse delivered education was less costly and more effective compared with no active intervention, producing a gain in QALYs (0.06 per participant) and cost savings of £27 per participant (9).”

4. A lot of information is provided about other studies but no information is given in comparison to the study performed in the manuscript. The section needs to be rewritten so that the relation between previous research and the submitted study is discussed. Now it is a lot of information not only in regard to comment 3 but also in regard to other previous studies.

This section has been rewritten explaining the need for more research in habit formation weight loss and commercial programmes.

“It is evident from existing literature that GPs play a crucial role in obesity prevention and weight management and are gate keepers to lifestyle weight management programmes. While there is evidence to suggest GP prescribed commercial programmes and/or weight-loss education is effective, further research is needed to explore the relationship between habit-formation programmes

such as 10TT with commercial programmes, with the aim to determine what the long-term cost savings and QALYs produced potentially would be over a long-time horizon.”

Acknowledgements

The first of this section is not really written to acknowledge but rather to inform more about the study. The author’s would benefit from rewriting it accordingly. However, it can be kept as it is.

No further comment

Tables

1. N refers to the full population. You are using a sample of the population in your tables so therefore should n be used and the headers in tables be updated accordingly.

N changed to “n” in all tables.

2. The reference to table 2 at row 218 is not in line with the content of it. A statement which is referring to the actual content of Table 2 is needed.

Text has been added in to describe the results.

“There were no differences between the two groups in terms of age, gender, prior costs and baseline EQ-5D scores (P= 0.47, 0.65, 0.29 and 0.41) (Table 2). Therefore, these were not included in the final model.”

3. In Table 4, as well as in abstract and results, two value digits is required used. Especially important for the incremental utility as it is now stated as 0.000 while it must be slightly above 0 considering that the NMB is above 0.

This has been changed throughout the paper to 0.001.

4. In Table 5, it is still not clear that discounted QALY’s refer to two years, which is my interpretation.

Discounted QALYs (24 months) inserted in table 5.

Cheers statement checklist

Non-important adjustments: i) 17 should be 9-11, ii) 22 should be 14-19 I assume and iii) 23 and 24 are only on page 20 (assuming that “Competing interests” corresponds to “Conflicts of interest”).

Cheers statement has been updated using the clean manuscript copy. See attachment (clean copy of the manuscript).

VERSION 4 – REVIEW

REVIEWER	Fredrik Norström Umeå University
REVIEW RETURNED	09-Apr-2018
GENERAL COMMENTS	Cost-effectiveness of habit-based advice for weight control

versus usual care in general practice in the Ten Top Tips (10TT) Trial: economic evaluation based on a randomised controlled trial

General comment from reviewer 2:

The revised version have handled the revisions well, and I only have minor revision and some grammatical errors that I would like to see improved.

Author's Response

General

1. There are still some linguistic errors in the text but it has been much improved. See e.g. row 223 where words are in wrong order, grammar at row 217, no space before counterweight at row 336, at row 352 are things messed up between sentences and row 378 need to be rephrased ("are" twice in it). I also doubt that "Except" is correctly used in footnotes of Table 4. I strongly advice that a native language speaker read through the whole text and strengthens the text.

• grammar at row 217

Changed to "Incremental costs were initially adjusted for; age, gender, practice and costs two years prior to implementation of 10TT. Adjusting for these factors did not make a significant difference to the results. Therefore, these were not included in the final model".

RESPONSE: Should there be a ";" in the sentence? It does not feel correct.

2. GLM vs OLS comment: Your explanation is ok for me. However, I recommend to improve the description so that it is clear for what independent variables the analysis is performed.

The following independent variables were used to predict the incremental costs and QALYs; age, gender, practice, prior costs from the previous two years and baseline utility values. Adjusting for these variables did not make statistically significant differences. P values were calculated for the independent variables, these were non-significant.

"A linear regression model using ordinary least squares (OLS) regression was used in the final analyses to estimate the mean difference in observed QALYs between the two trial arms. The independent variables used to perform the analysis were age, gender, practice and baseline utility values. There were no differences between the two groups in terms of age, practice, gender, prior costs and baseline EQ-5D scores (P= 0.47, 0.65, 0.29 and 0.41) (Table 2). Therefore, these were not included in the final model."

RESPONSE: The new sentence is very confusing. First, some variables are listed as independent variables and then it excludes some of these but also a few that was not listed previously. I assume that the problem is that the second sentence mixes independent variables with the outcome variable. Rewrite this sentence so that the actual message

	is correct. You are referring to Table 2 as reason for exclusion. It is fine but then you need to present the p-values for the tests that supposedly have been done in Table 2 to motivate why variables were excluded from further analyses. Discussion 1. You write “Our findings showed that weight loss programmes such as Weight Watchers lost more weight than education programmes and resulted in the possibility of cost savings, which we outline below”. This statement implies that results have been presented in regard to Weight Watchers in the manuscripts result section which is not the case. This sentence needs to be rewritten or there need to be something written about Weight Watchers (or similar) in results section. I suspect that the problem is that “Our findings” refer to what you find in previous publications and not to your own analyses. Then you need to rewrite the sentence so that this is clear. This section has been rewritten as suggested- “We found that that commercial weight loss programmes were highly prescribed amongst primary care providers and these participants lost more weight than those in self-led education programmes alone” RESPONSE: OK. You need to correct “that that” though in the sentence. Tables 3. In Table 4, as well as in abstract and results, two value digits is required used. Especially important for the incremental utility as it is now stated as 0.000 while it must be slightly above 0 considering that the NMB is above 0. This has been changed throughout the paper to 0.001. RESPONSE: OK. Note that a “)” is missing in the table’s first row “Base case*” and need to be corrected. 4. In Table 5, it is still not clear that discounted QALY’s refer to two years, which is my interpretation. Discounted QALYs (24 months) inserted in table 5. RESPONSE: OK. Note that “,” is missing in “Usual care” for Discounted QALYs without imputation.
--	---

VERSION 4 – AUTHOR RESPONSE

Author's Response

General

1. There are still some linguistic errors in the text but it has been much improved. See e.g. row 223 where words are in wrong order, grammar at row 217, no space before counterweight at row 336, at row 352 are things messed up between sentences and row 378 need to be rephrased (“are” twice in it). I also doubt that “Except” is correctly used in footnotes of Table 4. I strongly advice that a native language speaker read through the whole text and strengthens the text.

Nishma:

- Row 223: The sentence “Therefore, these were not included in the final model.” has been deleted.

- Row 217: This has been corrected- “;” the sentence has been rewritten (see below).

- Row 336: I’m unable to locate “Counterweight” at row 336. The only place I can find this word it at is row 312 and the spacing seems to be correct.

- Row 352: This sentence has been rewritten
“Finally, we were unable to access Hospital Episode Statistics (HES) data containing detailed secondary care resource use of NHS services by patients. Obtaining HES data would been problematic as this data would need to be linked to HES data by patient ID. Given the short timeframe of this trial, this was not feasible(35). Where data were available for secondary care and missing (inpatient admissions, A&E visits, outpatient clinic and other outpatient services), this was accounted for using multiple imputation, assuming these data were missing at random.”

- Row 378: This sentence has been rephrased- “Similarly, there was no significant difference in costs and QALYs between the two arms at 24 months.” I’m unable to locate the duplicate “are” you stated for this row.

- Except in footnotes of Table 4: This was deleted in the previous submission- I’m unable to locate “except” in Table 4.

• grammar at row 217

Changed to “Incremental costs were initially adjusted for; age, gender, practice and costs two years prior to implementation of 10TT. Adjusting for these factors did not make a significant difference to the results. Therefore, these were not included in the final model”.

RESPONSE: Should there be a “;” in the sentence? It does not feel correct.

Nishma: This has been rewritten as “Incremental costs were initially adjusted for; age, gender, practice and costs two years prior to implementation of 10TT. Adjusting for these factors did not make a significant difference to the results, and therefore not included in the final model.”

2. GLM vs OLS comment: Your explanation is ok for me. However, I recommend to improve the description so that it is clear for what independent variables the analysis is performed.

The following independent variables were used to predict the incremental costs and QALYs; age, gender, practice, prior costs from the previous two years and baseline utility values. Adjusting for these variables did not make statistically significant differences. P values were calculated for the independent variables, these were nonsignificant.

“A linear regression model using ordinary least squares (OLS) regression was used in the final analyses to estimate the mean difference in observed QALYs between the two trial arms. The independent variables used to perform the analysis were age, gender, practice and baseline utility values. There were no differences between the two groups in terms of age, practice, gender, prior costs and baseline EQ-5D scores (P= 0.47, 0.65, 0.29 and 0.41) (Table 2). Therefore, these were not included in the final model.”

RESPONSE: The new sentence is very confusing. First, some variables are listed as independent variables and then it excludes some of these but also a few that was not listed previously. I assume that the problem is that the second sentence mixes independent variables with the outcome variable. Rewrite this sentence so that the actual message is correct. You are referring to Table 2 as reason for exclusion. It is fine but then you need to present the p-values for the tests that supposedly have been done in Table 2 to motivate why variables were excluded from further analyses.

Nishma: This has now been rewritten- "We had initially adjusted for age, gender, practice and costs two years prior in the analysis for incremental costs. Similarly, we adjusted for age, gender, practice and baseline utility values in the analysis for incremental QALYs. There were no differences between the two groups in terms of these factors (Table 2) and therefore an unadjusted model was used."

Discussion

1. You write "Our findings showed that weight loss programmes such as Weight Watchers lost more weight than education programmes and resulted in the possibility of cost savings, which we outline below". This statement implies that results have been presented in regard to Weight Watchers in the manuscripts result section which is not the case. This sentence needs to be rewritten or there need to be something written about Weight Watchers (or similar) in results section. I suspect that the problem is that "Our findings" refer to what you find in previous publications and not to your own analyses. Then you need to rewrite the sentence so that this is clear.

This section has been rewritten as suggested- "We found that that commercial weight loss programmes were highly prescribed amongst primary care providers and these participants lost more weight than those in self-led education programmes alone"

RESPONSE: OK. You need to correct "that that" though in the sentence.

Nishma: Duplicate "that" removed from row 302.

Tables

3. In Table 4, as well as in abstract and results, two value digits is required used. Especially important for the incremental utility as it is now stated as 0.000 while it must be slightly above 0 considering that the NMB is above 0.

This has been changed throughout the paper to 0.001.

RESPONSE: OK. Note that a ")" is missing in the table's first row "Base case*" and need to be corrected.

Nishma: ")" inserted in Table 4 for Base case* (95% CI).

4. In Table 5, it is still not clear that discounted QALY's refer to two years, which is my interpretation.

Discounted QALYs (24 months) inserted in table 5.

RESPONSE: OK. Note that “,” is missing in “Usual care” for Discounted QALYs without imputation.

Nishma: “,” inserted in Table 5 Discounted QALYS (24 months) (95% CI).